**Characteristics of PM$_{2.5}$ mass concentrations and chemical species in urban and background**
**areas of China: emerging results from the CARE-China network**
Zirui Liu[1*], Wenkang Gao[1], Yangchun Yu[1], Bo Hu[1], Jinyuan Xin[1], Yang Sun[1], Lili Wang[1], Gehui
Wang[3], Xinhui Bi[4], Guohua Zhang[4], Honghui Xu[5], Zhiyuan Cong[6], Jun He[7], Jingsha Xu[7], Yuesi
Wang[1,2*]
[1]State Key Laboratory of Atmospheric Boundary Layer Physics and Atmospheric Chemistry, Institute of
Atmospheric Physics, Chinese Academy of Sciences, Beijing 100029, China
[2]Center for Excellence in Regional Atmospheric Environment, Institute of Urban Environment, Chinese Academy
of Sciences, Xiamen 361021, China
[3]State Key Laboratory of Loess and Quaternary Geology, Institute of Earth Environment, Chinese Academy of
Sciences, Xi'an 710075, China
[4]State Key Laboratory of Organic Geochemistry, Guangzhou Institute of Geochemistry, Chinese Academy of
Sciences, Guangzhou 510640, China
[5]Zhejiang Meteorology Science Institute, Hangzhou 310017, China
[6]Key Laboratory of Tibetan Environment Changes and Land Surface Processes, Institute of Tibetan Plateau Research,
Chinese Academy of Sciences, Beijing 100101, China
[7]International Doctoral Innovation Centre, The University of Nottingham Ningbo China, Ningbo 315100, China
[*]Corresponding author: Z.R Liu (Liuzirui@mail.iap.ac.cn); Y.S Wang (wys@mail.iap.ac.cn)
**Abstract:** The "Campaign on atmospheric Aerosol REsearch" network of China (CARE-China) is
a long-term project for the study of the spatiotemporal distributions of physical aerosol
characteristics as well as the chemical components and optical properties of aerosols over China.
This study presents the first long-term datasets from this project, including three years of
observations of online PM$_{2.5}$ mass concentrations (2012-2014) and one year of observations of PM$_{2.5}$
compositions (2012-2013) from the CARE-China network. The average PM$_{2.5}$ concentrations at 20
urban sites is 73.2 μg/m$^3$ (16.8-126.9 μg/m$^3$), which was three times higher than the average value
from the 12 background sites (11.2-46.5 μg/m$^3$). The PM$_{2.5}$ concentrations are generally higher in
east-central China than in the other parts of the country due to their relative large particulate matter
(PM) emissions and the unfavorable meteorological conditions for pollution dispersion. A distinct
seasonal variability of the PM$_{2.5}$ is observed, with highs in the winter and lows during the summer
at urban sites. Inconsistent seasonal trends were observed at the background sites. Bimodal and
unimodal diurnal variation patterns were identified at both urban and background sites. The
chemical compositions of PM$_{2.5}$ at six paired urban and background sites located within the most
polluted urban agglomerations (North China Plain (NCP), Yangtze River Delta (YRD), Pearl River
Delta (PRD), Northeast China Region (NECR), Southwestern China Region (SWCR)) and cleanest
regions (Tibetan Autonomous Region (TAR)) of China were analyzed. The major PM$_{2.5}$ constituents
across all the urban sites are organic matter (OM, 26.0%), SO$_4^{2-}$(17.7%), mineral dust (11.8%), NO$_3^-$
(9.8%), NH$_4^+$ (6.6%), elemental carbon (EC) (6.0%), Cl$^-$ (1.2%) at 45% RH and unaccounted matter
(20.7%). Similar chemical compositions of PM$_{2.5}$ were observed at background sites but were
associated with higher fractions of OM (33.2%) and lower fractions of NO$_3^-$ (8.6%) and EC (4.1%).
Significant variations of the chemical species were observed among the sites. At the urban sites, the
OM ranged from 12.6 μg/m$^3$ (Lhasa) to 23.3 μg/m$^3$ (Shenyang), the SO$_4^{2-}$ ranged from 0.8 μg/m$^3$

(Lhasa) to 19.7 μg/m$^3$ (Chongqing), the NO$_3^-$ ranged from 0.5 μg/m$^3$ (Lhasa) to 11.9 μg/m$^3$ (Shanghai) and the EC ranged from 1.4 μg/m$^3$ (Lhasa) to 7.1 μg/m$^3$ (Guangzhou). The PM$_{2.5}$ chemical species at the background sites exhibited larger spatial heterogeneities than those at urban sites, suggesting the different contributions from regional anthropogenic or natural emissions and from the long-range transport to background areas. Notable seasonal variations of PM$_{2.5}$ polluted days were observed, especially for the megacities in east-central China, resulting in frequent heavy pollution episodes occurring during the winter. The evolution of the PM$_{2.5}$ chemical compositions on polluted days was consistent for the urban and nearby background sites, where the sum of sulfate, nitrate and ammonia typically constituted much higher fractions (31-57%) of PM$_{2.5}$ mass, suggesting fine particle pollution in the most polluted areas of China assumes a regional tendency, and the importance to address the emission reduction of secondary aerosol precursors including SO$_2$ and NOx. Furthermore, distinct differences in the evolution of [NO$_3^-$]/[SO$_4^{2-}$] ratio and OC/EC ratio in polluted days imply that mobile sources and stationary (coal combustion) sources are likely more important in Guangzhou and Shenyang, respectively, whereas in Beijing it is mobile emission and residential sources. As for Chongqing, the higher oxidation capacity than the other three cities suggested it should pay more attention to the emission reduction of secondary aerosol precursors. This analysis reveals the spatial and seasonal variabilities of the urban and background aerosol concentrations on a national scale and provides insights into their sources, processes, and lifetimes.

## 1. Introduction

Atmospheric fine particulate matter (PM$_{2.5}$) is a complex heterogeneous mixture, whose physical size distribution and chemical composition change in time and space and are dependent on the emission sources, atmospheric chemistry, and meteorological conditions (Seinfeld and Pandis, 2016). Atmospheric PM$_{2.5}$ has known important environmental impacts related to visibility degradation and climate change. Because of their abilities to scatter and absorb solar radiation, aerosols degrade visibility in both remote and urban locations and can have direct and indirect effects on the climate (IPCC, 2013). Fine atmospheric particles are also a health concern and have been linked to respiratory and cardiovascular diseases (Sun et al., 2010; Viana et al., 2008; Zhang et al., 2014a). The magnitudes of the effects of PM$_{2.5}$ on all these systems depend on their sizes and chemical compositions. Highly reflective aerosols, such as sulfates and nitrates, result in direct cooling effects, while aerosols with low single-scattering albedos absorb solar radiation and include light-absorbing carbon, humic-like substances, and some components of mineral soils (Hoffer et al., 2006). The health impacts of these particles may also differ with different aerosol compositions (Zimmermann, 2015); the adverse health effects specifically associated with organic aerosols have been reported by Mauderly and Chow (2008). Therefore, the uncertainties surrounding the roles of aerosols in climate, visibility, and health studies can be significant because chemical composition data may not be available for large spatial and temporal ranges.

Reducing the uncertainties associated with aerosol effects requires observations of aerosol mass concentrations and chemical speciation from long-term spatially extensive ground-based networks. Continental sampling using ground-based networks has been conducted in North America (Hand et al., 2012) and Europe (Putaud et al., 2010) since the 1980s, such as via the U.S. EPA's Chemical Speciation Network (CSN), the Interagency Monitoring of Protected Visual Environments

(IMPROVE) network, the Clean Air Status and Trends Network (CASTNET) and the National
Atmospheric Deposition Program (NADP). Previous studies suggest the spatial and temporal
patterns of $PM_{2.5}$ mass concentrations and chemical species can vary significantly depending on
species and location. For example, Malm et al. (2004) reported the 2001 monthly mean speciated
aerosol concentrations from the IMPROVE monitors across the United States and demonstrated that
ammonium sulfate concentrations were highest in the eastern United States and dominated the fine
particle masses in the summer. Clearly decreasing gradients of the $SO_4^{2-}$ and $NO_3^-$ contributions to
$PM_{10}$ were observed in Europe when moving from rural to urban to kerbside sites (Putaud et al.,
2010). Although large disparities of $PM_{2.5}$ pollution levels exist between those megacities in
developing and developed countries, the $PM_{2.5}$ annual mass concentrations in the former are
approximately 10 times greater than those of the latter (Cheng et al., 2016); however, ground-based
networks that consistently measures $PM_{2.5}$ mass concentrations and chemical compositions remain
rare in the densely populated regions of developing countries.
China is the world's most populous country and has one of the fastest-growing economies. Fast
urbanization and industrialization can cause considerable increases in energy consumption. China's
energy consumption increased 120% from 2000 to 2010. Coal accounted for most of the primary
energy consumption (up to 70%) (Department of Energy Statistics, National Bureau of Statistics of
China, 2001; 2011). Meanwhile, the emissions of high concentrations of numerous air pollutants
cause severe air pollution and haze episodes. For example, a heavy air pollution episode occurred
in northeastern China in January of 2013, wherein the maximum hourly averaged $PM_{2.5}$ exceeded
600 $\mu gm^{-3}$ in Beijing (Wang et al., 2014). This event led to considerable public concern. However,
ground-based networks that consistently measure $PM_{2.5}$ mass concentrations and chemical
compositions in China are limited. Although there were some investigations of the various aerosol
chemical compositions in China (He et al., 2001; Huang et al., 2013; Li et al., 2012; Liu et al., 2015;
Pan et al., 2013; Tao et al., 2014; Wang et al., 2013; Yang et al., 2011; Zhao et al., 2013a; Zhou et
al., 2012), earlier studies were limited in their temporal and spatial scopes, with very few having
data exceeding one year while covering various urban and remote regions of the country (Zhang et
al., 2012; Wang et al., 2015b). Indeed, before 2013, the Chinese national monitoring network did
not report measurements of $PM_{2.5}$ or its chemical composition, and thus, ground-based networks for
atmospheric fine particulate matter measurements at regional and continental scales are needed as
these networks are essential for the development and implementation of effective air pollution
control strategies and are also useful for the evaluation of regional and global models and satellite
retrievals.
To meet these sampling needs, the "Campaign on atmospheric Aerosol REsearch" network of
China (CARE-China) was established in late 2011 for the study of the spatiotemporal distributions
of the physical and chemical characteristics and optical properties of aerosols (Xin et al., 2015).
This study presents the first long-term dataset to include three years of observations of online $PM_{2.5}$
mass concentrations (2012-2014) and one year of observations of $PM_{2.5}$ compositions (2012-2013)
from the CARE-China network. The purpose of this work is to (1) assess the $PM_{2.5}$ mass
concentration levels, including the seasonal and diurnal variation characteristics at the urban, rural
and regional background sites; to (2) obtain the seasonal variations of the $PM_{2.5}$ chemical
compositions at paired urban/background sites in the most polluted regions and clean areas; and to

(3) identify the occurrences and chemical signatures of haze events via an analysis of the temporal evolutions and chemical compositions of PM$_{2.5}$ on polluted days. These observations and analyses provide general pictures of atmospheric fine particulate matter in China and can also be used to validate model results and implement effective air pollution control strategies.

## 2 Materials and methods

### 2.1 An introduction to the PM$_{2.5}$ monitoring sites

The PM$_{2.5}$ data from 36 ground observation sites used in this study were obtained from the CARE-China network (Campaign on the atmospheric Aerosol REsearch network of China), which was supported by the Chinese Academy of Sciences (CAS) Strategic Priority Research Program grants (Category A). Xin et al. (2015) provided an overview of the CARE-China network, the cost-effective sampling methods employed and the post-sampling instrumental methods of analysis. Four more ground observation sites (Shijiazhuang, Tianjin, Ji'nan and Lin'an) from the "Forming Mechanism and Control Strategies of Haze in China" group (Wang et al., 2014) were also included in this study to better depict the spatial distributions and temporal variations of the PM$_{2.5}$ in eastern China. A comprehensive 3-year observational network campaign from 2012 to 2014 was carried out at these 40 ground observation sites. Figure 1 and Table 1, respectively, show the geographic distribution and details of the network stations, which include 20 urban sites, 12 background sites and 8 rural/suburban sites. The urban sites, such as those at Beijing, Shanghai and Guangzhou, are locations surrounded by typical residential areas and commercial districts. The background sites are located in natural reserve areas or scenic spots, which are far away from anthropogenic emissions and are less influenced by human activities. Rural/suburban sites are situated in rural and suburban areas, which may be affected by agricultural activities, vehicle emissions and some light industrial activities. These sites are located in different parts of China and can provide an integrated insight into the characteristic of PM$_{2.5}$ over China.

### 2.2 Online instruments and data sets

A tapered element oscillating microbalance (TEOM) was used for the PM$_{2.5}$ measurements at thirty-four sites within the network (Table S1). This system was designated by the US Environmental Protection Agency (USEPA) as having a monitoring compliance equivalent to the National Ambient Air Quality standard for particulate matter (Patashnick and Rupprecht 1991). The measurement ranges of the TEOMs were 0-5 g/m$^3$, with a 0.1 μg/m$^3$ resolution and precisions of ±1.5 (1-h average) and ±0.5 μg/m$^3$. The models used in the network are TEOM 1400a and TEOM 1405, and the entire system was heated to 50 °C; thus, a loss of semi-volatile compounds cannot be avoided. Our previous study showed that up to 25% lower mass concentrations were found for select daily means than those observed with gravimetric filter measurements, depending on the ammonium-nitrate levels and ambient temperatures (Liu et al., 2015). The errors of the TEOM measurements are systematic in that they are always negative. Thus, these errors may not be important for the study of the spatial distributions and temporal variations of PM$_{2.5}$. The other six sites of the network (Shanghai, Guangzhou, Chengdu, Xi'an, Urumchi and Qinghai Lake) were equipped with beta gauge instruments (EBAM, Met One Instruments Inc., Oregon). The measurement range of EBAM is 0-1000 μg/m$^3$, with a precision of 0.1 μg/m$^3$ and a resolution of 0.1 μg/m$^3$. The filters were changed every week, and the inlet was cleaned every month. The flow rates were also monitored and concurrently calibrated. A year-long intercomparison of daily PM$_{2.5}$ mass

concentrations measured by TEOM and EBAM was conducted at the Beijing site (Fig. S1a), and
the results showed that these two on-line instruments correlated well ($R^2$=0.90, P<0.01). TEOM
reported approximately 24% lower mass concentration than EBAM, and the difference could be
explained by the loss of semi-volatile materials from TEOM (Zhu et al., 2007).

**2.3 Filter sampling and chemical analysis**

In this study, filter sampling was conducted at the five urban sites of Beijing, Guangzhou, Lhasa,
Shenyang and Chongqing as well as at the six background sites of Xinglong, Lin'an, Dinghu
Mountain, Namsto, Changbai Mountain and Gongga Mountain. The Automatic Cartridge Collection
Unit (ACCU) system of Rupprecht & Patashnick Co. with 47 mm diameter quartz fiber filters (Pall
Life Sciences, Ann Arbor, MI, USA) was deployed in Beijing to collect the $PM_{2.5}$ samplers (Liu et
al., 2016a). Similar to the ACCU system, a standard 47 mm filter holder with quartz fiber filters
(Pall Life Sciences, Ann Arbor, MI, USA) was placed in the bypass line of TEOM 1400a and TEOM
1405 using quick-connect fittings and was used to collect the $PM_{2.5}$ samplers of the other nine sites,
excepting Guangzhou and Lin'an. Each set of the $PM_{2.5}$ samples was continuously collected over 48
h on the same days of each week, generally starting at 8:00 a.m. The flow rates were typically
15.6 L/min. For the Guangzhou site, the fine particles were collected on Whatman quartz fiber filters
using an Andersen model SA235 sampler (Andersen Instruments Inc.) with an air flow rate of
1.13 $m^3$/min. The sampling lasted 48h for the first three samples and 24 h for the rest samples,
generally starting at 8:00 a.m. For the Lin'an site, a medium volume $PM_{2.5}$ sampler (Model: TH-
150CIII, Tianhong Instrument CO., Ltd. Wuhan, China) was used to collect 24 h of $PM_{2.5}$ aerosols
on 90 mm quartz fiber filters (QMA, Whatman, UK) once every 6 days (Xu et al., 2017). The
sampling periods of these 11 urban and background sites are shown in Table S1.
All the filters were heat treated at 500 °C for at least 4 h for cleaning prior to filter sampling.
The $PM_{2.5}$ mass concentrations were obtained via the gravimetric method with an electronic balance
with a detection limit of 0.01 mg (Sartorius, Germany) after stabilizing at a constant temperature
(20±1 °C) and humidity (45%±5%). $PM_{2.5}$ mass concentrations measured by gravimetric method
correlated well with the on-line instruments (TEOM and EBAM) as showed in Fig. S1b. On average,
$PM_{2.5}$ mass concentrations measured by the filter sampling was approximately 9% higher than the
on-line instruments. Three types of chemical species were measured using the methods described in
Xin et al. (2015). Briefly, the organic carbon (OC) and elemental carbon (EC) values were
determined using a thermal/optical reflectance protocol using a DRI model 2001 carbon analyzer
(Atmoslytic, Inc., Calabasas, CA, USA) with the thermal/optical reflectance (TOR) method. A circle
piece of 0.495 $cm^2$ was cut off from the filters and was sent into the thermal optical carbon analyzer.
In a pure helium atmosphere, OC1, OC2, OC3 and OC4 are produced stepwise at 140 °C, 280 °C,
480 °C and 580 °C, respectively; followed by EC1 (540 °C), EC2 (780 °C) and EC3 (840 °C) in a
2% oxygen-contained helium atmosphere. Eight main ions, including $K^+$, $Ca^{2+}$, $Na^+$, $Mg^{2+}$, $NH_4^+$,
$SO_4^{2-}$, $NO_3^-$ and $Cl^-$, were measured via ion chromatography (using a Dionex DX 120 connected to
a DX AS50 autosampler for anions and a DX ICS90 connected to a DX AS40 autosampler for
cations). One-quarter of each filter substrate was extracted with 25 mL deionized water in a PET
vial for 30 min. Before performing a targeted sample analysis, a standard solution and blank test
were performed, and the correlation coefficient of the standard samples was more than 0.999. The
detection limits for all anions and cations, which were calculated as three times the standard
deviations of seven replicate blank samples, are all lower than 0.3 μg m$^{-3}$ (Liu et al., 2017). The
microwave acid digestion method was used to digest the filter samples into liquid solution for
elemental analysis. One quarter of each filter sample was placed in the digestion vessel with a
mixture of 6 mL $HNO_3$, 2 mL $H_2O_2$ and 0.6 mL HF, and was then exposed to a three-stage
microwave digestion procedure from a microwave-accelerated reaction system (MARS, CEM
Corporation, USA). After that, 18 elements, including Mg, Al, K, Ca, V, Cr, Mn, Fe, Co, Ni, Cu, Zn,
As, Se, Ag, Cd, Tl and Pb, were determined by Agilent 7500a inductively coupled plasma mass
spectrometry (ICP-MS, Agilent Technologies, Tokyo, Japan). Quantification was carried out by the
external calibration technique using a set of external calibration standards (Agilent Corporation) at
concentration levels close to that of the samples. The relative standard deviation for each
measurement (repeated twice) was within 3%. The method detection limits (MDLs) were
determined by adding 3 standard deviations of the blank readings to the average blank values (Yang
et al., 2009). Quality control and quality assurance procedures were routinely applied for all the
carbonaceous, ion and elemental analysis.

**3. Results and discussions**
**3.1 Characteristics of PM$_{2.5}$ mass concentrations at urban and background sites**
**3.1.1 Average PM$_{2.5}$ levels**

The location, station information and average PM$_{2.5}$ concentrations from the 40 monitoring
stations are shown in Fig. 1 and Table 1. The highest PM$_{2.5}$ concentrations were observed at the
urban stations of Xi'an (125.8 μg/m$^3$), Taiyuan (111.5 μg/m$^3$), Ji'nan (107.5 μg/m$^3$) and Shijiazhuang
(105.1 μg/m$^3$), which are located in the most polluted areas of the Guanzhong Plain (GZP) and the
North China Plain (NCP). Several studies have revealed that the enhanced PM$_{2.5}$ pollutions of the
GZP and NCP are not only due to the primary emissions from local sources such as the local
industrial, domestic and agricultural sources but are also due to secondary productions (Huang et
al., 2014; Guo et al., 2014; Wang et al., 2014). Furthermore, the climates of the GZP and NCP are
characterized by stagnant weather with weak winds and relatively low boundary layer heights,
leading to favorable atmospheric conditions for the accumulation, formation and processing of
aerosols (Chan and Yao, 2008). Note that the averaged PM$_{2.5}$ concentrations in Beijing and Tianjin
were approximately 70 μg/m$^3$, which is much lower than those of the other cities, including Ji'nan
and Shijiazhuang in the NCP, possibly because Beijing and Tianjin are located in the northern part
of the NCP, far from the intense industrial emission area that is mainly located in the southern part
of the NCP. Interestingly, the average PM$_{2.5}$ concentrations at Yucheng (102.8 μg/m$^3$) and Xianghe
(83.7 μg/m$^3$) were even higher than most of those from the urban stations. Although Yucheng is a
rural site, it is located in an area with rapid urbanization near Ji'nan and is therefore subjected to the
associated large quantities of air pollutants. In addition, Xianghe is located between Beijing and
Tianjin and is influenced by the regionally transported contributions from nearby megacities and
the primary emissions from local sources. Yantai is a coastal city with relatively low PM
concentrations compared to those of with inland cities on the NCP.
The PM$_{2.5}$ concentrations were also high in the Yangtze River Delta (YRD), which is another
developed and highly-populated city cluster area like the NCP (Fu et al., 2013). The average PM$_{2.5}$
values of the three urban stations of Shanghai, Wuxi and Hefei were 56.2, 65.2 and 80.4 μg/m$^3$,
respectively, which are comparable to those of the megacities of Beijing and Tianjin in the NCP.
Due to the presence of fewer coal-based industries and dispersive weather conditions, the $PM_{2.5}$
concentrations of the Pearl River Delta (PRD) are generally lower than those of the other two largest
city clusters in China, such as those from the NCP and YRD. The average $PM_{2.5}$ value at Guangzhou
was 44.1 $μg/m^3$, which was similar to the $PM_{2.5}$ values of the background stations from the NCP and
YRD. Shenyang, the capital of the province of Liaoning, is located in the Northeast China Region
(NECR), which is an established industrial area. High concentrations of trace gases and aerosol
scattering in the free troposphere have been observed via aircraft observations and are due to
regional transports and heavy local industrial emissions (Dickerson et al., 2007). In the present study,
the average $PM_{2.5}$ concentration of Shenyang was 77.6 $μg/m^3$. Meanwhile, Hailun, which is a rural
site in northeastern China, had an average $PM_{2.5}$ concentration of 41.6 $μg/m^3$, which was much
lower than that of the rural site of Yucheng in the NCP.
High aerosol optical depths and low visibilities have been observed in the Sichuan Basin
(Zhang et al., 2012), which is located in the Southwestern China Region (SWCR). The poor
dispersion conditions and heavy local industrial emissions make this another highly polluted area in
China. In the present study, the average $PM_{2.5}$ concentration in Chengdu was measured as 102.2
$μg/m^3$, which is much higher than the averages from the megacities of Beijing, Shanghai and
Guangzhou but is comparable to those of Ji'nan and Shijiazhuang. Chongqing, another megacity
located in the SWCR, however, showed much lower $PM_{2.5}$ values than Chengdu. Urumqi, the capital
of the Uighur Autonomous Region of Xinjiang, located in northwestern China, experiences air
pollution due to its increasing consumption of fossil fuel energy and steadily growing fleet of motor
vehicles (Mamtimin and Meixner, 2011). The average $PM_{2.5}$ concentration measured in Urumqi is
104.1 $μg/m^3$, which is comparable to those of the urban sites in the GZP and NCP. The similarity
among the $PM_{2.5}$ values for Cele, Dunhuang and Fukang is due to their location, being far from
regions with intensive economic development but strongly affected by sandstorms and dust storms
due to their proximity to dust source areas. For example, the average $PM_{2.5}$ concentration in Cele
during the spring (200.7 $μg/m^3$) was much greater than those of the other three seasons. Lhasa, the
capital of the Tibet Autonomous Region (TAR), is located in the center of the Tibetan Plateau at a
very high altitude of 3700 m. The $PM_{2.5}$ concentrations in Lhasa were low, with average values of
30.6 $μg/m^3$, because of its relatively small population and few industrial emissions.
Much lower $PM_{2.5}$ concentrations were observed at the background stations, the values of
which ranged from 11.2 to 46.5 $μg/m^3$. The lowest concentration of $PM_{2.5}$ was observed in Namsto,
a background station on the TAR with nearly no anthropogenic effects. The highest $PM_{2.5}$
concentration of the background stations was observed at Lin'an, a background station in the PRD.
The average $PM_{2.5}$ concentration at the urban and background sites in this study are shown as box-
plots in Fig. S2a. The average $PM_{2.5}$ concentration of the background stations (a total of 12 sites) is
28.5 $μg/m^3$, and the average concentration of the $PM_{2.5}$ values from urban stations (a total of 20
sites) is 73.2$μg/m^3$. The latter value is approximately three times the former, suggesting the large
differences in fine particle pollution at urban and background sites across China. To further
characterize these kinds of differences for different parts of China, six pairs of $PM_{2.5}$ values
measured from urban and background stations were selected to represent the NCP, YRD, PRD, TAR,
NECR and SWCR, respectively (Fig. S2). The first three areas (NCP, YRD and PRD) and the last
two areas (NECR and SWCR) were the most industrialized and populated regions in China, while
TAR is the cleanest area in China. The $PM_{2.5}$ concentrations of the background stations in the NCP,
YRD and PRD are 39.8 μg/m$^3$ (Xinglong), 46.5 μg/m$^3$ (Lin'an) and 40.1 μg/m$^3$ (Dinghu Mountain)
and are much higher than those of the background stations in other parts of China, which are usually
below 25 μg/m$^3$. All values especially for those observed in urban and rural sites in this study were
much greater than the results from Europe and North America. For urban/suburban sites, average
$PM_{2.5}$ concentrations of 20.1 μg/m$^3$ was reported by Gehrig and Buchmann (2003) from 1998 to
2001 in Switzerland, and average concentrations of 16.3 μg/m$^3$ for the period 2008-2009 in the
Netherlands (Janssen et al., 2013). Between October 2008 and April 2011, the 20 study areas
covered major cities of the European ESCAPE project showed annual average concentrations of
$PM_{2.5}$ ranging from 8.5 to 29.3 μg/m$^3$, with low concentrations in northern Europe and high
concentrations in southern and eastern Europe (Eeftens et al., 2012). Based on a constructed
database of $PM_{2.5}$ component concentrations from 187 counties in the United States for 2000-2005,
Bell et al. (2007) reported an average $PM_{2.5}$ value of 14.0 μg/m$^3$, with higher values in the eastern
United States and California, and lowest values in the central regions and Northwest. For
background sites, Putaud et al. (2010) showed that annual average of $PM_{2.5}$ ranged from 3 to
22μg/m$^3$ observed from 12 background sites across Europe. In addition, average $PM_{2.5}$ value of
12.6μg/m$^3$ was observed at a regional background site in the Western Mediterranean from 2002 to
2010 (Cusack et al., 2012).
**3.1.2 Seasonal variations of $PM_{2.5}$ mass concentrations**
Generally, the $PM_{2.5}$ concentrations in urban areas show distinct seasonal variabilities, with
maxima during the winter and minima during the summer for most of China (Fig. 1), which is a
similar pattern to that of the results reported by Zhang and Cao (2015). In northern and northeastern
China, the wintertime peak values of $PM_{2.5}$ were mainly attributed to the combustion of fossil fuels
and biomass burning for domestic heating over extensive areas, which emit large quantities of
primary particulates as well as the precursors of secondary particles (He et al., 2001). In addition,
new particle formation and the secondary production of both inorganic aerosols and OM could
further enhance fine PM abundance (Huang et al., 2014; Guo et al., 2014). Furthermore, the
planetary boundary layer is relatively low in the winter, and more frequent occurrences of stagnant
weather and intensive temperature inversions cause very bad diffusion conditions, which can result
in the accumulation of atmospheric particulates and lead to high-concentration PM episodes (Quan
et al., 2014; Zhao et al., 2013b). In southern and eastern China, although the effect of domestic
heating is not as important as that in northern China, the weakened diffusion and transport of
pollutants from the north due to the activity of the East Asian Winter Monsoon reinforces the
pollution from large local emissions in the winter more than in any other season (Li et al., 2011;
Mao et al., 2017). For northwestern and West Central China, the most polluted season is the spring
instead of the winter due to the increased contribution from dust particles in this desert-like region
(Zou and Zhai, 2004), suggesting that the current $PM_{2.5}$ control strategies (i.e., reducing fossil/non-
fossil combustion derived VOCs and PM emissions) will only partly reduce the $PM_{2.5}$ pollution in
western of China. $PM_{2.5}$ is greatly decreased during the summer in urban areas, which is associated
with the reduced anthropogenic emissions from fossil fuel combustion and biomass burning
domestic heating. Further, the more intense solar radiation causes a higher atmospheric mixing layer,
which leads to strong vertical and horizontal aerosol dilution effects (Xia et al., 2006). In addition,
increased precipitation in most of China due to the summer monsoon can increase the wet
scavenging of atmospheric particles. As a result, $PM_{2.5}$ minima are observed in the summer at urban
sites.
The seasonal variations of $PM_{2.5}$ at the background sites varied in different parts of China
(Fig. 3). Dinghu Mountain and Lin'an showed maximum values in the winter, while Zangdongnan,
Qinghai Lake, Xishuangbanna and Mount Everest showed maximum values in the spring. In
addition, a summer maximum of $PM_{2.5}$ was observed for Xinglong, and an autumn maximum was
observed for Tongyu. Changbai Mountain, Gongga Mountain and Namsto showed weak seasonal
variabilities. These results suggest the different contributions from regional anthropogenic and
natural emissions and long-range transports to background stations. The monthly average $PM_{2.5}$
concentrations of the urban and background sites in the NCP, YRD, PRD, TAR, NECR and SWCR
are further analyzed and shown in Fig. 2. The monthly variations of the $PM_{2.5}$ concentrations at the
background sites in the YRD and PRD were consistent with those of the nearby urban sites, both of
which showed maximum values in December (YRD) and January (PRD). The reasons for this
similarity are primarily the seasonal fluctuations of emissions, which are already well known due to
the similar variations of other parameters, including sulfur dioxide and nitrogen oxide, as shown in
Fig. S3. In contrast, the monthly variations of $PM_{2.5}$ at Xinglong showed different trends than those
of the nearby urban stations. The maximum value of $PM_{2.5}$ at this site was observed in July, while
the maximum value in Beijing was observed in January. The reasons for this are not primarily the
seasonal fluctuations of emissions, but rather meteorological effects (frequent inversions during the
winter and strong vertical mixing during the summer). The Xinglong site is situated at an altitude of
900 m a.s.l., and therefore, during the wintertime, the majority of cases above the inversion layer
are protected from the emissions of the urban agglomerations of the NCP. Furthermore, in the NCP
area, northerly winds prevail in the winter, while southerly winds prevail in the summer. Thus, in
the summer, more air masses from the southern urban agglomerations will lead to high $PM_{2.5}$
concentrations in Xinglong. Weak monthly variabilities were observed for Namsto, Changbai
Mountain and Gongga Mountain, although remarkable monthly variabilities were found at the
nearby cities of Lhasa, Shenyang and Chongqing. The reasons for this difference are mainly that
these three sites are elevated remote stations that are far from human activities and show
predominant meteorological influences.
**3.1.3 Diurnal variations of $PM_{2.5}$ mass concentrations**
To derive importance information to identify the potential emission sources and the times
when the pollution levels exceed the proposed standards, hourly data were used to examine the
diurnal variabilities of $PM_{2.5}$ as well as those of the other major air pollutants. Fig. 3 illustrates the
diurnal variations of the hourly $PM_{2.5}$ concentrations in Beijing, Shanghai, Guangzhou, Lhasa,
Shenyang and Chongqing, in the largest megacities in the NCP, YRD, PRD, TAR, NECR and SWCR
and in the different climatic zones of China, respectively. Of the urban sites, Lhasa has the lowest
$PM_{2.5}$ concentrations, but the most significant pronounced diurnal variations of $PM_{2.5}$, with obvious
morning and evening peaks appearing at 10:00 and 22:00 (Beijing Time) due to the contributions of
enhanced anthropogenic activity during the rush hours. The minimum value occurred at 16:00,
which is mainly due to a higher atmospheric mixing layer, which is beneficial for air pollution

diffusion. This bimodal pattern was also observed in Shenyang and Chongqing, which show morning peaks at 7:00 and 9:00 and evening peaks at 19:00 and 20:00, respectively. However, the $PM_{2.5}$ values in Beijing, Shanghai and Guangzhou showed much weaker urban diurnal variation patterns, and slightly higher $PM_{2.5}$ concentrations during the night than during the day were observed, which can be explained by the enhanced emissions from heating and the relatively low boundary layer. Moreover, fine particles emitted from diesel truck traffic which is allowed only during nighttime would additionally increase $PM_{2.5}$ burden because emission factors of heavy-duty vehicles are 6 times than those from light-duty vehicles (Westerdahl et al., 2009). Note that the morning peaks in Beijing, Shanghai and Guangzhou were not as obvious as those of other cities, although both the $SO_2$ and $NO_2$ values increased due to increased anthropogenic emissions (Fig. S4). Alternatively, this decreasing trend may be the result of an increasing boundary layer depth. The invisible morning peak of $PM_{2.5}$ in these three cities was possibly attributed to the stricter emission standards applied at recently years. As showed in Fig.S5, the morning peak of $PM_{2.5}$ in Beijing was gradually disappeared or invisible after National 5 vehicle emission standard applied at the beginning of 2013 (www.bjpc.gov.cn). The same thing would be also observed in Shanghai and Guangzhou which implemented the same vehicle emission standards followed Beijing, while it not true for the other cities as the latest vehicle emission standard was usually applied 2-3 years later than the three megacities. At the urban sites of Beijing, Shanghai and Guangzhou, the $PM_{2.5}$ levels started to increase in the late afternoon, which could be explained by the increasing motor vehicle emissions as $NO_2$ is also dramatically increased during the same period.

At the background area of the TAR, significant pronounced diurnal variations of $PM_{2.5}$ were observed in Namsto, with a morning peak at 9:00 and an evening peak at 21:00 (Fig. 3d), which are similar to those of the urban site of Lhasa. As there are hardly any anthropogenic activities near Namsto, this kind of diurnal pattern of $PM_{2.5}$ may be influenced by the evolution of the planetary boundary layer. Both Gongga Mountain and Lin'an showed the same bimodal pattern of $PM_{2.5}$ as that in Namsto, the former site could also be influenced by the planetary boundary layer, while the latter site was not only influenced by the evolution of the planetary boundary layer but also would be highly affected by the regional transportation from the YRD region. For the background site of the NCP, however, Xinglong showed smooth $PM_{2.5}$ variations. As mentioned before, the Xinglong station is located on the mountain and has an altitude of 960 m a.s.l. The mixed boundary layer of the urban area increases in height in the morning and reaches a height of approximately 1000 meters in the early afternoon. Then, the air pollutants from the urban area start to affect the station as the vertical diffusion of the airflow and the $PM_{2.5}$ concentration reach their maxima at 18:00. Next, the concentration starts to decrease when the mixed boundary layer collapses in the late afternoon, eventually forming the nocturnal boundary layer (Boyouk et al., 2010). Thus, $PM_{2.5}$ concentration decreased slowly during the night and morning, reaching a minimum at 10:00. At Dinghu Mountain and Changbai Mountain, the daytime $PM_{2.5}$ greater than that of the nighttime, with a maximum value occurring at approximately 11:00-12:00. This kind of diurnal pattern of $PM_{2.5}$ is mainly determined by the effects of the mountain-valley breeze. Both the Dinghu Mountain and Changbai Mountain stations are located near the mountain. Thus, during daytime, the valley breeze from urban areas carries air pollutants that will accumulate in front of the mountain and cause an increase of the PM concentration. Meanwhile, at night, the fresh air carried by the mountain breeze will lead to the

dilution of the PM, so low concentrations are sustained during the night. Further support for this
pattern comes from the much higher maximum values of $PM_{2.5}$ in the winter than those in the
summer, as enhanced air pollutant emissions in urban areas are expected in the winter due to heating.
**3.2 Chemical compositions of $PM_{2.5}$ in urban and background sites**
**3.2.1 Overview of $PM_{2.5}$ mass speciation**

Figure 4 shows the annual average and seasonal average chemical compositions of $PM_{2.5}$ at

six urban and six background sites, which represent the largest megacities and regional background
areas of the NCP, YRD, PRD, TAR, NECR and SWCR. The chemical species of $PM_{2.5}$ in Shanghai
were obtained from Zhao et al. (2015). The atmospheric concentrations of the main $PM_{2.5}$
constituents are also shown in Table 2. The EC, nitrate ($NO_3^-$), sulfate ($SO_4^{2-}$), ammonium ($NH_4^+$)
and chlorine ($Cl^-$) concentrations were derived directly from measurements. Organic matter (OM)
was calculated assuming an average molecular weight per carbon weight, showing an OC of 1.6 at
the urban sites and of 2.1 at the background sites, based on the work of Turpin and Lim (2001);
however, these values are also spatially and temporally variable, and typical values could range from
1.3 to 2.16 (Xing, et al., 2013). The calculation of mineral dust was performed on the basis of crustal
element oxides ($Al_2O_3$, $SiO_2$, $CaO$, $Fe_2O_3$, $MnO_2$ and $K_2O$). In addition, the Si content, which was
not measured in this study, was calculated based on its ratio to Al in crustal materials (Mason, 1966);
namely, $[Si]=3.41\times[Al]$. Finally, the unaccounted-for mass refers to the difference between the
$PM_{2.5}$ gravimetric mass and the sum of the PM constituents mentioned above.

The PM constituents' relative contributions to the PM mass are independent of their dilutions

and reflect differences in the sources and processes controlling the aerosol compositions (Putaud et
al., 2010). When all the main aerosol components except water are quantified, they account for 73.6-
84.8% of the $PM_{2.5}$ mass (average 79.2%) at urban sites and for 76.2-91.1% of the $PM_{2.5}$ mass
(average 83.4%) at background sites. The remaining unaccounted-for mass fraction may be the
result of analytical errors, a systematic underestimation of the PM constituents whose concentrations
are calculated from the measured data (e.g., OM, and mineral dust), and aerosol-bound water
(especially when mass concentrations are determined at RH >30%). For the urban sites, the mean
composition given in descending concentrations is 26.0% OM, 17.7% $SO_4^{2-}$, 11.8% mineral dust,
9.8% $NO_3^-$, 6.6% $NH_4^+$, 6.0% EC and 1.2% $Cl^-$. For the background sites, the mean composition
given in descending concentrations is 33.2% OM, 17.8% $SO_4^{2-}$, 10.1% mineral dust, 8.7% $NH_4^+$,
8.6% $NO_3^-$, 4.1% EC and 0.9% $Cl^-$. Generally, the chemical compositions of the $PM_{2.5}$ at background
sites are similar to those of the urban sites, although they show a much higher fraction of OM and
lower fractions of $NO_3^-$ and EC. Significant seasonal variations of the chemical compositions were
observed at urban sites (Fig. 4c), with much higher fractions of OM (33.7%) and $NO_3^-$ (11.1%) in
the winter and much lower fractions of OM (20.7%) and $NO_3^-$ (6.9%) in the summer. In contrast,
the fraction of $SO_4^{2-}$ was consistent among the different seasons, although its absolute concentration
in the winter (14.9 μg/m$^3$) was higher than that in the summer (11.7 μg/m$^3$). Compared with those
at urban sites, different seasonal variation of OM were observed at the background sites, which
showed summer maxima and winter/spring minima (Fig. 4d). While the wintertime peaks of OM at
the urban sites were probably due to additional local emissions sources related to processes like
heating, the summer peaks at the background sites were attributed to the enhanced biogenic
emissions. Note that the seasonal variations of $NO_3^-$ were similar to those at urban sites; this seasonal
phenomenon is due to the favorable conditions of cold temperature and high relative humidity
conditions leading to the formation of particulate nitrate. The seasonal behaviors of $SO_4^{2-}$ at the
background sites were markedly different than those of the urban sites and indicate very different
sources and atmospheric processing of $SO_4^{2-}$, which will be further discussed for specific regions of
China.

There are significant variations of the absolute speciation concentrations at these urban and

background sites (Table 2). For the urban sites, the OM concentrations span a 2-fold concentration
range from 12.6 $\mu g/m^3$ (Lhasa) to 23.3 $\mu g/m^3$ (Shenyang), while these values range from 3.4 $\mu g/m^3$
(Namtso) to 21.7 $\mu g/m^3$ (Lin'an) at the background sites. The $SO_4^{2-}$ and $NO_3^-$ concentrations exhibit
larger spatial heterogeneities than those of the OM for both urban and background sites. The
absolute values of $SO_4^{2-}$ have an approximately 25-fold range in urban sites, from 0.8 $\mu g/m^3$ (Lhasa)
to 19.7 $\mu g/m^3$ (Chongqing), while this value has a 30-fold range at the background sites, from 0.4
$\mu g/m^3$ (Namsto) to 11.2 $\mu g/m^3$ (Lin'an). The corresponding mass fractions are 26.8% in Chongqing
and below 3% in Lhasa. Much higher fractions of $SO_4^{2-}$ in the $PM_{2.5}$ were observed at the urban sites
located in southern China than those in northern China, although the average concentration of $PM_{2.5}$
is greater in the north than in the south, suggesting that sulfur pollution remains a problem for
southern China (Liu, et al., 2016b). This problem may be attributed to higher sulfur contents of the
coal in southern China, with 0.51% in the north vs. 1.32% in the south and up to >3.5% in Chongqing
in southern China (Lu et al., 2010; Zhang et al., 2010). In addition, the higher fraction of sulfate in
south China is also likely associated to the higher oxidation capacity in south China and therefore
higher formation efficiency from $SO_2$ to $SO_4^{2-}$. The absolute values of $NO_3^-$ have an approximately
20-fold range in urban sites and a greater than 100-fold range in background sites. This
heterogeneity reflects the large spatial and temporal variations of the NO$x$ sources. For the urban
sites, the absolute EC values have a 5-fold concentration range, from 1.4 $\mu g/m^3$ (Lhasa) to greater
than 7.0 $\mu g/m^3$ (Guangzhou), while this species has a 15-fold concentration range at the background
sites and is mainly from anthropogenic sources. In comparison, the absolute concentrations of
mineral dust exhibit much weaker spatial variations at the urban and background sites.
The characteristics of the $PM_{2.5}$ chemical compositions at individual site were discussed in
more detail. In this section, six pairs of urban and background sites from each region of China were
selected, and the differences in the chemical compositions of urban and background sites were
analyzed.
**3.2.2 North China Plain**
Beijing is the capital of China and has attracted considerable attention due to its air pollution
(Chen et al., 2013). Beijing is the largest megacity in the NCP, which is surrounded by the Yanshan
Mountains to the west, north and northeast and is connected to the Great North China Plain to the
south. The filter sampler is located in the courtyard of the Institute of Atmospheric Physics (IAP)
(116.37°E, 39.97°N), 8 km northwest of the center of downtown. The $PM_{2.5}$ concentration during
the filter sampling period was 71.7 $\mu g/m^3$, which is close to the three-year average $PM_{2.5}$ value
reported by TEOM (Table 1). $PM_{2.5}$ in Beijing is mainly composed by OM (26.6%), $SO_4^{2-}$ (16.5%)
and $NO_3^-$ (13.0%) (Fig. 5a), which compare well with previous studies (Yang et al., 2011; Oanh et
al., 2006). However, the mineral dust fraction found in this study (6.5%) was much lower than that
found in Yang et al. (2011) (19%) but was comparable to that found in Oanh et al. (2006) (5%),
potentially due to difference in definitions. In addition, the EC fraction (5.7%) was slightly lower
than those found in previous studies (7%-7.4%) (Yang et al., 2011; Wang et al., 2015a). The annual
concentration of OM (19.1 μg/m$^3$) in Beijing was comparable to those in Shanghai, Guangzhou and
Chongqing, but was much lower than that in Shenyang. Higher fractions of OM were observed in
the winter (34.2%) and autumn (30.5%) than in the summer (21.6%) and spring (20.9%). The annual
concentration of SO$_4^{2-}$ (11.9 μg/m$^3$) was much lower than those of earlier years (15.8 μg/m$^3$, 2005-
2006) (Yang et al., 2011), suggesting that the energy structure adjustment implemented in Beijing
(e.g., replacing coal fuel with natural gas) has been effective in decreasing the particulate sulfate in
Beijing. Further support for this comes from the SO$_4^{2-}$ concentration in the winter (16.5 μg/m$^3$) being
comparable to that in the summer (13.4 μg/m$^3$). The significant NO$_3^-$ value (9.3 μg/m$^3$) reflects the
significant urban NOx emissions in Beijing, which was greatest during the winter, as expected from
ammonium-nitrate thermodynamics. The greater mineral component in the spring reflects the
regional natural dust sources.
The filter sampling site in Xinglong (117.58°E, 40.39°N) was located at Xinglong Observatory,
National Astronomical Observatory, Chinese Academy of Sciences, which is 110 km northeast of
Beijing (Fig. 1). This site is surrounded by mountains and is minimally affected by anthropogenic
activities. The PM$_{2.5}$ concentration during the filter sampling period was 42.6 μg/m$^3$, which is close
to the three-year average PM$_{2.5}$ values reported by TEOM (Table 1). The annual chemical
composition of the PM$_{2.5}$ in Xinglong was similar to that in Beijing, although relatively higher
fractions of OM and sulfate were observed in Xinglong (Fig. 5a). Higher fractions of OM were
found in the winter (36.7%), and higher fractions of sulfate were found in the summer (32.1%) than
in any other season (OM: 23.0-30.4%; SO$_4^{2-}$: 15.7-20.1%). Interestingly, the summer SO$_4^{2-}$
concentration in Xinglong (14.4 μg/m$^3$) was even higher than that in Beijing, suggesting spatially
uniform distributions of SO$_4^{2-}$ concentrations across the NCP. This result indicates that regional
transport can be an important source of SO$_4^2$ aerosols in Beijing, especially during the summer.
**3.2.3 Yangtze River Delta**
Shanghai is the economic center of China, lying on the edge of the broad flat alluvial plain of
the YRD, with a few mountains to the southwest. The filter sampler was located at the top of a four-
floor building of the East China University of Science and Technology (121.52°E, 31.15°N) (Zhao
et al., 2015), approximately 10 km northwest of the center of downtown. The PM$_{2.5}$ concentration
during the filter sampling period was 68.4 μg/m$^3$, which is greater than the three-year average PM$_{2.5}$
value reported by EBAM, likely due to the different sampling period (Table S1). The PM$_{2.5}$ in
Shanghai mainly comprises OM (24.9%), SO$_4^{2-}$ (19.9%) and NO$_3^-$ (17.4%), which is comparable to
the results of previous studies (Ye et al., 2003; Wang et al., 2016). This site had the highest NO$_3^-$
(11.9 μg/m$^3$) and the second-highest SO$_4^{2-}$ (13.6 μg/m$^3$) values of the urban sites, while its OM (17.1
μg/m$^3$) was comparable to those of Guangzhou and Chongqing. The SO$_4^{2-}$ and NO$_3^-$ values were
highest during the autumn as expected based on the widespread biomass burning in the autumn in
the YRD (Niu et al., 2013). However, the OM values were highest during the winter and mainly
originated from secondary aerosol processes based on the highest OC/EC ratios (6.0) and the poor
relationship of the OC and EC in this season.
Filter sampling was conducted at the Lin'an Regional Atmospheric Background Station
(119.73°E, 30.30°N), which is a background monitoring station for the World Meteorological
Organization (WMO) global atmospheric observation network. The Lin'an site was located at the
outskirts of Lin'an County within Hangzhou Municipality, which was 200 km southwest of
Shanghai (Fig. 1). This site is surrounded by agricultural fields and woods and is less affected by
urban, industrial and vehicular emissions (Xu et al., 2017). The $PM_{2.5}$ concentration during the filter
sampling period was 66.3 μg/m$^3$, which is higher than the three-year average $PM_{2.5}$ values reported
by TEOM, likely due to the different sampling period (Table S1). The annual chemical composition
of the $PM_{2.5}$ in Lin'an was different than that in Shanghai, with much higher fractions of OM (32.7%)
and $NH_4^+$ (11.0%). Furthermore, the absolute concentration of OM in Lin'an was much higher than
that in Shanghai, especially in the summer (21.7 vs. 9.9 μg/m$^3$), which may be attributed to the
enhanced biomass burning at both local and regional scales as well as the higher concentration of
summer EC in Lin'an than in Shanghai (2.2 vs. 1.4 μg/m$^3$). In addition, the $SO_4^{2-}$ and $NO_3^-$
concentrations in Lin'an were comparable to those in Shanghai. These results suggest a spatially
homogeneous distribution of secondary aerosols over the PRD and the the transportation of aged
aerosol and gas pollutants from city clusters has significantly changed the aerosol chemistry in the
background area of this region.

**3.2.4 Pearl River Delta**

Guangzhou is the biggest megacity in south China located in the PRD and mainly consists of
floodplains within the transitional zone of the East Asian monsoon system (Yang et al., 2011). The
filter sampler was set up on the rooftop of a 15-m high building of the Guangzhou Institute of
Geochemistry, Chinese Academy of Sciences (113.35°E, 23.12°N). This site was surrounded by
heavily trafficked roads and dense residential areas, representing a typical urban location. The $PM_{2.5}$
concentration during the filter sampling period was 75.3 μg/m$^3$, which is much higher than the three-
year average $PM_{2.5}$ value reported by EBAM (Table 1), likely due to the different sampling period
and location. The $PM_{2.5}$ in Guangzhou mainly comprises OM (22.2%), $SO_4^{2-}$ (17.3%) and mineral
dust (9.7%), which have values comparable to previous studies conducted in the years of 2013-2014
(Chen et al., 2016; Tao et al., 2017). This site has the lowest OC/EC ratio (1.5) of all urban sites,
which can be explained by the abundance of diesel engine truck in Guangzhou City (Verma et al.,
2010). Obvious seasonal variations of OM, $SO_4^{2-}$ and $NO_3^-$ were observed, showing winter/autumn
maxima and summer/spring minima. In addition, summer minima were also observed for EC and
$NH_4^+$. High mixing heights in the summer and clean air masses affected by summer monsoons from
the South China Sea should lead to the minima of these species in summer, while the low wind
speeds, weak solar radiation, relatively low precipitation (Tao et al., 2014) and relatively high
emissions (Zheng et al., 2009) result in the much higher concentrations of OM and secondary
inorganic aerosols ($SO_4^{2-}$, $NO_3^-$ and $NH_4^+$) in the winter and autumn.
Filter sampling was conducted at Dinghu Mountain Station (112.50°E, 23.15°N), which is
located in the middle of Guangdong Province in southern China. This site was surrounded by hills
and valleys, being approximately 70 km west of Guangzhou (Fig. 1). The $PM_{2.5}$ concentration
during the filter sampling period was 40.1 μg/m$^3$, close to the three-year average $PM_{2.5}$ values
reported by TEOM. Distinct seasonal variations of OM, $SO_4^{2-}$, $NO_3^-$ and $NH_4^+$ were observed, with
the highest concentration of OM and $NO_3^-$ occurring in the winter, while the highest concentrations
of $SO_4^{2-}$ and $NH_4^+$ occurred in the autumn. In contrast, EC and mineral dust showed weak seasonal
variations. Dinghu Mountain has the second-highest EC and $SO_4^{2-}$ values of the background sites,
being 2.0 μg/m$^3$ and 10.1 μg/m$^3$. In addition, the lowest OC/EC ratio was observed at Dinghu
Mountain (2.8); the other background sites had values ranging from 3.5-8.3. These results indicate
that this background site is intensely influenced by vehicular traffic, fossil fuel combustion and
industrial emissions due to the advanced urban agglomeration in the PRD region. These results are
consistent with the finds from previous studies (Liu et al., 2011; Wu et al., 2016). Compared with
those from Guangzhou, higher fractions of $SO_4^{2-}$ and $NO_3^-$ were observed at Dinghu Mountain,
while the fractions of OM and mineral dust were similar at these two sites, possibly indicating that
there was a significantly larger fraction of transported secondary aerosols or aged aerosols at the
background site of the PRD.
**3.2.5 Tibetan Autonomous Region**
Located in the inland TAR, Lhasa is one of the highest cities in the world (at an altitude of
3700 m). The city of Lhasa is located in a narrow west-east oriented valley in the southern part of
the TAR. The filter sampler was located on the roof of a 20-m high building on the campus of the
Institute of Tibetan Plateau Research (Lhasa branch) (91.63°E, 29.63°N). This site is close to Jinzhu
road, one of the busiest roads in the city (Cong et al., 2011). The PM$_{2.5}$ concentration during the
filter sampling period was 36.4 μg/m$^3$, which is close to the three-year average PM$_{2.5}$ values reported
by TEOM. The PM$_{2.5}$ in Lhasa mainly comprises OM (34.5%) and mineral dust (31.9%), and the
secondary inorganic aerosols ($SO_4^{2-,}$ $NO_3^-$ and $NH_4^+$) contributed little to the PM$_{2.5}$ (<5%). These
results are comparable to those of a previous study conducted in the year of 2013-2014 (Wan et al.,
2016). In addition, this site reports the lowest OM (12.6 μg/m$^3$), secondary inorganic aerosols (1.7
μg/m$^3$) and EC (1.4 μg/m$^3$) values of the urban sites in this study. Higher fractions of OM were
observed in the winter (48.4%) and spring (43.1%), exceeding those in the summer (24.6%) and
autumn (31.2%). Weak seasonal variations were found for the $SO_4^{2-}$ (1.5-3.0%) and $NO_3^-$ (1.1-1.7%)
values, suggesting the negligible contributions from fossil fuel combustion in Lhasa.
Filter sampling was conducted at the Namtso Monitoring and Research Station for Multisphere
Interactions (90.98°E, 30.77°N), a remote site located on the northern slope of the Nyainqen-tanglha
Mountains, approximately 125 km northwest of Lhasa (Fig. 1). The PM$_{2.5}$ concentration during the
filter sampling period was 9.5 μg/m$^3$, which is close to the three-year average PM$_{2.5}$ value reported
by TEOM. The PM$_{2.5}$ in Namtso mainly comprises mineral dust (40.8%) and OM (36.3%), while
$SO_4^{2-}$ and $NO_3^-$ contributed less than 5% to the PM$_{2.5}$. This chemical composition is distinctly
different from those of the other background sites in this study, but is comparable to the background
site at Qinghai Lake in the TAR (Zhang et al., 2014b). Namtso has the lowest OM, EC, $SO_4^{2-}$, $NO_3^-$
and $NH_4^+$ values of all the background sites in this study. Spring maxima and winter minima were
observed for the OM and EC, while the $SO_4^{2-}$, $NO_3^-$ and $NH_4^+$ values showed weak seasonal
variations. The highest OC/EC ratio was observed (8.3) at this site, suggesting that the organic
aerosols at Namtso mainly originated from secondary aerosol processes or aged organic aerosols
from regional transports.
**3.2.6 Northeast China Region**
Shenyang is the capital city of Liaoning province and the largest city in northeastern China.
The main urban area is located on a delta to the north of the Hun River. The filter sampler was
located at the Shenyang Ecological Experimental Station of the Chinese Academy of Science
(123.40°E, 41.50°N) and was surrounded by residential areas with no obvious industrial pollution
sources around the monitoring station, representing the urban area of Shenyang. The $PM_{2.5}$
concentration during the filter sampling period was 81.8 μg/m$^3$, which is close to the three-year
average $PM_{2.5}$ value reported by TEOM (Table 1). The $PM_{2.5}$ in Shenyang mainly comprises OM
(28.5%), $SO_4^{2-}$ (16.1%) and mineral dust (11.3%). This site reports the highest OM (23.3 μg/m$^3$) and
mineral dust (9.2 μg/m$^3$) values as well as the second-highest EC (5.2 μg/m$^3$) value of the urban
sites. The $NO_3^-$ concentration at this site, however, was the second-lowest of the urban sites (Table
2). Much higher fractions of OM were observed in the winter (40.5%) than in the other seasons
(15.6-26.5%) (Fig. 5), possibly due to the enhanced coal burning for winter heating. Further support
for this pattern comes from the high abundance of chlorine during the cold seasons, which is mainly
associated with coal combustion. The contribution from sea-salt particles is not important since the
sampling sites are at least 200 km from the sea. Note that the fraction of $SO_4^{2-}$ in the $PM_{2.5}$ during
the winter was lower than that in the summer, although the absolute concentration was much higher
in the winter (23.6 μg/m$^3$) than in the summer (11.3 μg/m$^3$). This result may be attributed to the
reduced transformation of sulfur dioxide at low temperatures.

Filter sampling was conducted at the Changbai Mountain forest ecosystem station (128.01°E,

42.40°N), which was mostly surrounded by hills and forest and is located approximately 390 km
northeast of Shenyang (Fig. 1). This site is situated 10 km from the nearest town, Erdaobaihe, which
has approximately 45000 residents. The sources of PM were expected to be non-local. Hence, this
site is considered a background site in the NECR. The $PM_{2.5}$ concentration during the filter sampling
period was 23.3 μg/m$^3$, which is close to the three-year average $PM_{2.5}$ value reported by TEOM
(Table 1). The main contributions to the $PM_{2.5}$ at Changbai Mountain were OM (38.1%), mineral
dust (16.0%) and $SO_4^{2-}$ (14.3%), similar to those in Shenyang. Note that the summer OM
concentrations were quite similar at these two sites (8.0 vs. 9.0 μg/m$^3$), but the OC/EC ratios were
different (4.8 vs. 1.6), which may reflect the different origins of the OM at the urban (primary
emissions) and background sites (secondary processes) of the NECR. The OM concentrations in the
other seasons were much lower at Changbai Mountain than those from Shenyang city, especially
during the winter (10.8 vs. 59.4 μg/m$^3$). In fact, weak seasonal variations of chemical species (OM,
EC, $SO_4^{2-}$, $NO_3^-$ and $NH_4^+$) were observed at Changbai Mountain. This site reports the second-
lowest values of OM, EC, $SO_4^{2-}$ and $Cl^-$ of the background sites. These results suggest that aerosols
at Changbai Mountain were influenced by the regional transports alone.
**3.2.7 Southwestern China Region**

Chongqing is the fourth municipality near Central China, lying on the Yangtze River in

mountainous southwestern China, near the eastern border of the Sichuan Basin and the western
border of Central China. For topographic reasons, Chongqing has some of the lowest wind speeds
in China (annual averages of 0.9-1.6 m s$^{-1}$ from 1979 to 2007; Chongqing Municipal Bureau of
Statistics, 2008), which favors the accumulation of pollutants. The filter sampler was located on the
rooftop of a 15-m high building on the campus of the Southwest University (106.54°E, 29.59°N).
This site is located in an urban district of Chongqing with no obvious industrial pollution sources
around the monitoring site, representing the urban area of Chongqing. The $PM_{2.5}$ concentration
during the filter sampling period was 73.5 μg/m$^3$, of which 26.8% is $SO_4^{2-}$, 23.5% OM, 10.0%
mineral dust, 8.9% $NO_3^-$, 8.2% EC and 6.5% $NH_4^+$. The OM fraction is smaller than those measured
by Yang et al. (2011) (32.7%) and Chen et al., 2017 (30.8%), while the $SO_4^{2-}$ fraction is greater than

673 the values reported in these two studies (19.8-23.0%). This site shows the highest $SO_4^{2-}$ (19.7 μg/m$^3$),

674 the highest $NH_4^+$ (6.1 μg/m$^3$) and the third-highest EC (4.8 μg/m$^3$) values of the urban sites. A weak

675 seasonal variation in the chemical composition of PM$_{2.5}$ was observed, although a much higher

676 concentration of this species was found in the winter than in the other seasons.

677   Filter sampling was performed at the Gongga Mountain Forest Ecosystem Research Station

678 (101.98°E, 29.51°N) in the Hailuogou Scenic Area, a remote site located in southeastern Ganzi in

679 the Tibetan Autonomous Prefecture in Sichuan province. This site is mostly surrounded by glaciers

680 and forests and is located approximately 450 km northwest of Chongqing (Fig. 1). The PM$_{2.5}$

681 concentration during the filter sampling period was 32.2 μg/m$^3$, close to the three-year average PM$_{2.5}$

682 value reported by TEOM (Table 1). The dominant components of PM$_{2.5}$ were OM (40.7%), $SO_4^{2-}$

683 (14.6%) and mineral dust (9.8%), similar to those at Changbai Mountain. This site has the second-

684 highest OM (13.1 μg/m$^3$) value of the background sites, which may mainly be due to secondary

685 processes, considering the high OC/EC ratio (5.6). In addition, distinct seasonal variations of OM

686 were observed, which shows summer maxima (19.9 μg/m$^3$) and autumn minima (9.1 μg/m$^3$).

687 Previous studies showed higher mixing ratios of the VOCs during the spring and summer and lower

688 mixing ratios during the autumn at Gongga Mountain (Zhang et al., 2014c), which may result in

689 high concentrations of OM in the summer because the OC/EC ratio reaches its highest value in the

690 summer (10.3). Second-lowest EC and $NO_3^-$ values of the background sites were observed here,

691 suggesting the insignificant influence of human activities in this region.

692 **3.3 Temporal evolution and chemical composition PM$_{2.5}$ in polluted days**

693 **3.3.1 Temporal evolution of PM$_{2.5}$ mass concentration in polluted days**

694   Using the "Ambient Air Quality Standard" (GB3095-2012) of China (CAAQS), the

695 occurrences of polluted days exceeding the daily threshold values during 2012-2014 were counted

696 for each site (Fig. 6). Based on the number of polluted days exceeding the CAAQS daily guideline

697 of 35 μg/m$^3$, substandard days of PM$_{2.5}$ account for more than 60% of the total period at the majority

698 of urban sites, excepting Lhasa, Taipei and Sanya. Note that the ten most polluted cities (Ji'nan,

699 Chengdu, Taiyuan, Hefei, Shenyang, Xi'an, Changsha, Shijiazhuang, Wuxi and Chongqing)

700 experienced less than 20% clean days (daily PM$_{2.5}$<35 μg/m$^3$) during the three-year observation

701 period. Interestingly, the occurrences of heavily polluted days (daily PM$_{2.5}$>150 μg/m$^3$) were

702 different among these ten most polluted cities. While more than 15% of the total period comprised

703 heavily polluted days in Ji'nan, Taiyuan, Chengdu, Xi'an and Shijiazhuang, heavily polluted days

704 accounted for less than 5% of the total days in the other five cities, which mainly experienced

705 slightly polluted (35-75 μg/m$^3$) and moderately polluted (75-115 μg/m$^3$) days. Due to the regional

706 pollutant transports, the rural and background sites near the most polluted cities also showed high

707 occurrences of polluted days. Polluted days accounted for more than 50% of the total period at

708 Xin'long, Lin'an and Dinghu Mountain. In addition, an even higher occurrence of polluted days

709 (>80%) was found for the rural areas of Yucheng and Xianghe. In contrast, the background sites in

710 the TAR, NECR and SWCR rarely experienced polluted days, and over 80% of the total period

711 comprised clean days at these sites.

712   The polluted days were not equally distributed throughout the year. The monthly distributions

713 for the polluted days at each site are shown in Fig. 7. In terms of the occurrences of heavily polluted

714 days, December, January and February were predominant months for the urban sites located in the

most polluted areas of the GZP and NCP, where both the unfavorable dispersion conditions for
pollutants and the additional emission enhancements from residential heating contributed to the
heavy pollution in the winter. The heavy pollution occurring in April and November in Cele was
primarily caused by sandstorms and dust storms. Heavily polluted days were rarely observed at the
12 background sites in this study. The moderately polluted and polluted days were still mainly
concentrated in the winter in the megacities of the GZP and NCP and also occurred in the winter in
the megacities of the YRD and SWCR. In addition, March to June and September to October were
periods with high occurrences of polluted days. Dust storms from northern China (March to April),
biomass burning after crop harvests (May to June and September to October) and worsening
dispersion conditions after the summers likely accounted for the polluted days (Cheng et al., 2014;
Fu et al., 2014). The majority of slightly polluted days occurred from June to September, except at
several urban sites in southern China. The mass level of 35-75 $\mu g/m^3$ was considered a low level of
pollution for the entire year, illustrating that the summer and early autumn experienced cleaner
conditions.
**3.3.2 Chemical evolution of PM$_{2.5}$ composition in polluted days**
The mean percentile compositions of the major components in PM$_{2.5}$ at different pollution
levels from four paired urban-background sites are shown in Fig. 8. With the pollution level
increased from clean to moderately polluted, the EC fraction in Beijing decreased slightly, the OM
fraction decreased significantly, and the sulfate and nitrate contributions increased sharply (Fig. 8a).
The same chemical evolution of the PM$_{2.5}$ was also observed at the background site of Xinglong,
suggesting that regional transport plays a vital role in the formation of the slightly and moderately
polluted days in the NCP. When the pollution level increased to heavily polluted, however, the OM
fraction further increased and was accompanied by increases of the sulfate and nitrate contributions
as well as decreases of the mineral dust contribution, indicating the enhanced secondary
transformation of gaseous pollutants (etc. SO$_2$, NO$x$, VOCs) during heavily polluted periods (Liu et
al., 2016a). Note that a steady increase of [NO$_3^-$]/[SO$_4^{2-}$] ratio was observed with the aggravation of
pollution (Fig. 8a), suggesting the relatively more important contribution of mobile than stationary
sources (Arimoto et al., 1996). In addition, much higher OC/EC ratios were found in Beijing,
especially during the heavily polluted days (OC/EC=6.3) (Fig. 8), compared with Guangzhou,
Shenyang and Chongqing. Higher OC/EC ratio has been reported to be emitted from coal
combustion (2.7) and biomass burning (6.6) than from motor vehicles (1.1) (Watson et al., 2001;
Saarikoski et al., 2008). In the Northern China, the residential sector is the largest emitter of
carbonaceous aerosols (Lei et al., 2011; Lu et al., 2011), which are formed by the inefficient
combustion of fossil fuel and biomass in unregulated cooking and heating devices. For OC, the
residential sector contribution can exceed 95% (Liu, et al., 2016c). Thus, the highest OC/EC ratio
in Beijing indicates that residential emissions would also contributed considerably to the
development of heavily polluted days.
Unlike in Beijing, the contributions of OM and EC were almost constant across the different
pollution levels in Guangzhou, while the contribution of the secondary inorganic aerosols (SIA)
increased slightly (Fig. 8b). Interestingly, the nitrate contribution increased faster than that of the
sulfate when the pollution level increased from clean to heavily polluted, similar to the patterns of
Beijing. Furthermore, the [NO$_3^-$]/[SO$_4^{2-}$] ratio increased continuously and it reported the highest
ratio of $[NO_3^-]/[SO_4^{2-}]$ (1.3) during the heavily polluted days in Guangzhou (Fig. 8). At the same
time, the ratio of OC/EC was nearly constant with the aggravation of pollution, and it reported the
lowest OC/EC ratio (1.6-1.8) among the four megacities. These results suggest the dominate
contribution of local traffic emissions in the development of fine particulate pollution. The chemical
evolution of $PM_{2.5}$ at the background site of PRD was similar to that of the urban site at Guangzhou,
although a significant contribution of SIA was observed when the pollution level increased from
clean to moderately polluted (34% vs. 58%). Note that the contribution of sulfate increased sharply,
suggesting that regional transports dominated the particle pollution during heavily polluted days.

Compared with Beijing, a reversed chemical evolution of $PM_{2.5}$ for the different pollution

levels was observed in Shenyang, with the OM fraction increasing sharply from 22% to 37%, while
the SIA decreased slightly from 39% to 31% (Fig. 8c). Note that a steady increase of sulfate from
slightly polluted days to heavily polluted days was observed. In addition, a nearly constant low ratio
of $[NO_3^-]/[SO_4^{2-}]$ (0.30-0.38) and continually increased ratio of OC/EC (2.3-4.5) was observed with
the aggravation of pollution. These results suggest that enhanced local stationary emissions like coal
combustion dominate the temporal evolution of $PM_{2.5}$ on polluted days in Shenyang. The highest
concentration of $Cl^-$ in Shenyang than other cities in this study further support the significant
contribution of coal combustion. A similar chemical evolution of $PM_{2.5}$ was found at the background
site of Changbai Mountain, which showed a significantly increased OM fraction and slightly
decrease of SIA when the pollution level increased from clean to slighted polluted, indicating the
enhanced contribution from local emissions like coal combustion for heating during slightly polluted
days. Further support for this pattern is seen in the increase of the EC fraction (Fig. 8 g).

Similar to that in Guangzhou, the contribution of OM was almost constant for different

pollution levels in Chongqing, while much higher contribution of SIA was observed, especially
during the heavily polluted days. In addition, a steady increase of $[NO_3^-]/[SO_4^{2-}]$ ratio was observed,
similar with those in Beijing and Guangzhou, suggesting the relatively more important contribution
of mobile than stationary sources (Arimoto et al., 1996). Furthermore, the OC/EC ratio was also
continually increased with the aggravation of pollution, and different from that in Guangzhou but
similar with that in Shenyang. Note that the fraction of OM, sulfate and nitrate during the heavily
polluted days in Chongqing was much higher than those in Beijing, Guangzhou and Shenyang,
suggesting the higher oxidation capacity and therefore higher formation efficiency from gaseous
pollutants (etc. $SO_2$, NO$x$, VOCs) to secondary aerosol. These results suggest the importance of
local traffic emissions and the formation of secondary aerosol in driving $PM_{2.5}$ pollution in
Chongqing. The background site of Gongga Mountain shows decreased contributions of OM, EC,
SIA and mineral dust when the pollution level increased from clean to slightly polluted days, similar
to the pattern observed in Xinglong. Note that the unaccounted-for fraction was largely increased
on slightly polluted days (33% vs. 10%), possibly due to the increase of aerosol-bound water related
to the hygroscopic growth of aerosols at high RH values on slightly polluted days (Bian et al., 2014).
**4. Conclusions**

We have established a national-level network ("Campaign on atmospheric Aerosol REsearch"

network of China (CARE-China)) that conducted continuous monitoring of $PM_{2.5}$ mass
concentrations at 40 ground observation station, including 20 urban sites, 12 background sites and
8 rural/suburban sites. The average aerosol chemical composition was inferred from the filter
samples from six paired urban and background sites, which represent the largest megacities and
regional background areas in the five most polluted regions and the TAR of China. This study
presents the first long-term dataset including three-year observations of online $PM_{2.5}$ mass
concentrations (2012-2014) and one year observations of $PM_{2.5}$ compositions (2012-2013) from the
CARE-China network. One of the major purposes of this study was to compare and contrast urban
and background aerosol concentrations from nearby regions. The major findings include the
following:
(1) The average $PM_{2.5}$ concentration from 20 urban sites is 73.2 μg/m$^3$ (16.8-126.9 μg/m$^3$), which
is three times greater than the average value of 12 background sites (11.2-46.5 μg/m$^3$). The highest
$PM_{2.5}$ concentrations were observed at the stations on the Guanzhong Plain (GZP) and the NCP. The
$PM_{2.5}$ pollution is also a serious problem for the industrial regions of northeastern China and the
Sichuan Basin and is a relatively less serious problem for the YRD and the PRD. The background
$PM_{2.5}$ concentrations of the NCP, YRD and PRD were comparable to those of the nearby urban sites,
especially for the PRD. A distinct seasonal variability of the $PM_{2.5}$ is observed, presenting peaks
during the winter and minima during the summer at the urban sites, while the seasonal variations of
$PM_{2.5}$ at the background sites vary in different part of China. Bimodal and unimodal diurnal
variation patterns were identified at both the urban and background stations.
(2) The major $PM_{2.5}$ constituents across all the urban sites are OM (26.0%), $SO_4^{2-}$ (17.7%),
mineral dust (11.8%), $NO_3^-$ (9.8%), $NH_4^+$ (6.6%), EC (6.0%), $Cl^-$ (1.2%) at 45% RH and
unaccounted matter (20.7%). Similar chemical compositions of $PM_{2.5}$ were observed for the
background sites and were associated with higher fractions of OM (33.2%) and lower fractions of
$NO_3^-$ (8.6%) and EC (4.1%). Analysis of filter samples reveals that several $PM_{2.5}$ chemical
components varied by more than an order of magnitude between sites. For urban sites, the OM
ranges from 12.6 μg/m$^3$ (Lhasa) to 23.3 μg/m$^3$ (Shenyang), the $SO_4^{2-}$ ranges from 0.8 μg/m$^3$ (Lhasa)
to 19.7 μg/m$^3$ (Chongqing), the $NO_3^-$ ranges from 0.5 μg/m$^3$ (Lhasa) to 11.9 μg/m$^3$ (Shanghai) and
the EC ranges from 1.4 μg/m$^3$ (Lhasa) to 7.1 μg/m$^3$ (Guangzhou). The $PM_{2.5}$ chemical species of
the background sites exhibit larger spatial heterogeneities than those of the urban sites, suggesting
the different contributions from regional anthropogenic and natural emissions and from the long-
range transport to background areas.
(3) Notable seasonal variations of $PM_{2.5}$ polluted days were observed, especially for the
megacities in east-central China, resulting in frequent heavy pollution episodes occurring during the
winter. The increasing contribution of secondary aerosol on polluted days was observed both for the
urban and nearby background sites, suggesting fine particle pollution in the most polluted areas of
China assumes a regional tendency, and the importance to address the emission reduction of
secondary aerosol precursors. In addition, the chemical species dominating the evolutions of the
heavily polluted events were different, while decreasing or constantly contribution of OM associated
with increasing contribution of SIA characteristic evolution of $PM_{2.5}$ in NCP, PRD and SWCR, the
opposite phenomenon was observed in NECR. Further analysis from the $[NO_3^-]/[SO_4^{2-}]$ ratio and
OC/EC ratio showed that fine particle pollution in Guangzhou and Shenyang was mainly attributed
to the traffic emissions and coal combustion, respectively, while more complex and variable major
sources including mobile vehicle emission and residential sources contributed to the development
of heavily polluted days in Beijing. As for Chongqing, the higher oxidation capacity than other cities
suggested it should pay more attention to the emission reduction of secondary aerosol precursors.
These results suggest the different formation mechanisms of the heavy pollution in the most polluted
city clusters, and unique mitigation measures should be developed for the different regions of China.
The seasonal and spatial patterns of urban and background aerosols emphasize the importance
of understanding the variabilities of the concentrations of major aerosol species and their
contributions to the $PM_{2.5}$ budget. Comparisons of $PM_{2.5}$ chemical compositions from urban and
background sites of adjacent regions provided meaningful insights into aerosol sources and transport
and into the role of urban influences on nearby rural regions. The integration of data from 40 sites
from the CARE-China network provided an extensive spatial coverage of fine particle
concentrations near the surface and could be used to validate model results and implement effective
air pollution control strategies.

Acknowledgments
This study was supported by the Ministry of Science and Technology of China (Grant nos.
2017YFC0210000), the National Natural Science Foundation of China (Grant nos. 41705110) and
the Strategic Priority Research Program of the Chinese Academy of Sciences (Grant nos.
XDB05020200 & XDA05100100). We acknowledge the tremendous efforts of all the scientists and
technicians involved in the many aspects of the Campaign on atmospheric Aerosol REsearch
network of China (CARE-China).

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

Table 1 Geographic information and three-year mean PM$_{2.5}$ concentration of the monitor stations.

| Station/Code | Latitude, Longitude | Altitude(m) | Station type | Mean(µg/m3) | N(day) |
|---|---|---|---|---|---|
| Beijing/BJC | 39.97°N, 116.37°E | 45 | Northern city | 69.4±54.8 | 1077 |
| Cele/CLD | 37.00°N, 80.72°E | 1306 | Northwestern country | 126.9±155.4 | 600 |
| Changbai Mountain/CBM | 42.40°N, 128.01°E | 738 | Northeastern background | 17.6±12.6 | 807 |
| Changsha/CSC | 28.21°N, 113.06°E | 45 | Central city | 77.9±45.4 | 1045 |
| Chengdu/CDC | 30.67°N, 104.06°E | 506 | Southwestern city | 102.2±66.2 | 1008 |
| Chongqing/CQC | 29.59°N, 106.54°E | 259 | Southwestern city | 65.1±35.8 | 972 |
| Dinghu Mountain/DHM | 23.17°N, 112.50°E | 90 | Pearl River Delta background | 40.1±25.0 | 954 |
| Dunhuang/DHD | 40.13°N, 94.71°E | 1139 | Desert town | 86.2±94.3 | 726 |
| Fukang/FKZ | 44.28°N, 87.92°E | 460 | Northwestern country | 69.9±69.6 | 960 |
| Gongga Mountain/GGM | 29.51°N, 101.98°E | 1640 | Southwestern background | 25.5±15.5 | 869 |
| Guangzhou/GZC | 23.16°N, 113.23°E | 43 | Southern city | 44.1±23.8 | 772 |
| Hailun/HLA | 47.43°N, 126.63°E | 236 | Northeastern country | 41.6±45.0 | 1076 |
| Hefei/HFC | 31.86°N, 117.27°E | 24 | Eastern city | 80.4±45.3 | 909 |
| Ji'nan/JNC | 36.65°N, 117.00°E | 70 | Northern city | 107.8±57.4 | 701 |
| Kunming/KMC | 25.04°N, 102.73°E | 1895 | Southwestern city | 47.0±25.2 | 967 |
| Lhasa/LSZ | 29.67°N, 91.33°E | 3700 | Tibet city | 30.6±21.3 | 600 |
| Lin'an/LAZ | 30.30°N, 119.73°E | 139 | Eastern background | 46.5±27.2 | 1086 |
| Mount Everest/ZFM | 28.21°N, 86.56°E | 4700 | Tibet background | 24.4±25.1 | 390 |
| Namtso/NMT | 30.77°N, 90.98°E | 4700 | Tibet background | 11.2±6.9 | 499 |
| Nagri/ALZ | 32.52°N, 79.89°E | 4300 | Tibet background | 19.5±12.4 | 72 |
| Qianyanzhou/QYZ | 26.75°N, 115.07°E | 76 | Southeastern country | 52.1±28.4 | 927 |
| Qinghai Lake/QHL | 37.62°N, 101.32°E | 3280 | Tibet background | 16.2±17.0 | 590 |
| Sanya/SYB | 18.22°N, 109.47°E | 8 | Southern island city | 16.8±13.1 | 595 |
| Shanghai/SHC | 31.22°N, 121.48°E | 9 | Eastern city | 56.2±59.4 | 822 |
| Shapotou/SPD | 37.45°N, 104.95°E | 1350 | Desert background | 51.1±33.3 | 1016 |
| Shenyang/SYC | 41.50°N, 123.40°E | 49 | Northeastern city | 77.6±41.2 | 926 |
| Shijiazhuang/SJZ | 38.03°N, 114.53°E | 70 | Northern city | 105.1±92.7 | 1031 |
| Taipei/TBC | 25.03°N, 121.90°E | 150 | Island city | 22.1±10.7 | 1083 |
| Taiyuan/TYC | 37.87°N, 112.53°E | 784 | Northern city | 111.5±74.9 | 987 |
| Tianjin/TJC | 39.08°N, 117.21°E | 9 | Northern city | 69.9±49.6 | 1034 |
| Tongyu/TYZ | 44.42°N, 122.87°E | 160 | Inner Mongolia background | 24.5±24.5 | 757 |
| Urumchi/URC | 43.77°N, 87.68°E | 918 | Northwestern city | 104.1±145.2 | 776 |
| Wuxi/WXC | 31.50°N, 120.35°E | 5 | Eastern city | 65.2±36.8 | 1003 |
| Xi'An/XAC | 34.27°N, 108.95°E | 397 | Central city | 125.8±108.2 | 1077 |
| Xianghe/XHZ | 39.76°N, 116.95°E | 25 | North China suburbs | 83.7±62.3 | 1084 |
| Xinglong/XLZ | 40.40°N, 117.58°E | 900 | North China background | 39.8±34.0 | 1035 |
| Xishuangbanna/BNF | 21.90°N, 101.27°E | 560 | Southwestern rain forest | 25.0±18.7 | 707 |
| Yantai/YTZ | 36.05°N, 120.27°E | 47 | East China sea coast city | 51.1±36.7 | 915 |
| Yucheng/YCA | 36.95°N, 116.60°E | 22 | North China country | 102.8±61.8 | 1008 |
| Zangdongnan/ZDN | 29.77°N, 94.73°E | 2800 | Southern Tibet forest | 12.3±8.0 | 475 |

 Table 2 Summary of the concentrations of $PM_{2.5}$ and its components (μg/m$^3$) in urban and
 background sites.

| Station | $PM_{2.5}$ | OM | EC | $NO_3^-$ | $SO_4^{2-}$ | $NH_4^+$ | $MD^*$ | $Cl^-$ | Unaccounted$^{**}$ |
|---|---|---|---|---|---|---|---|---|---|
| Urban sites | | | | | | | | | |
| BJC(n=88) | 71.7(36.0) | 19.1(11.0) | 4.1(1.1) | 9.3(7.5) | 11.9(8.2) | 5.3(2.7) | 4.7(2.9) | 0.7(1.0) | 16.5(11.8) |
| SHC(n=120) | 68.4(20.3) | 17.1(4.5) | 2.0(0.6) | 11.9(5.0) | 13.6(6.4) | 5.8(2.1) | | | 18.1(4.9) |
| GZC(n=106) | 75.3(37.7) | 16.7(10.0) | 7.1(4.8) | 7.2(7.9) | 13.1(7.9) | 4.8(3.5) | 7.3(3.3) | 1.0(1.1) | 18.1(13.1) |
| LSZ(n=60) | 36.4(18.7) | 12.6(1.9) | 1.4(0.6) | 0.5(0.2) | 0.8(0.4) | 0.4(0.2) | 11.6(12.9) | 0.3(0.1) | 8.8(7.8) |
| SYC(n=36) | 81.8(55.6) | 23.3(22.3) | 5.2(3.4) | 4.6(4.7) | 13.2(10.7) | 4.5(2.6) | 9.2(5.6) | 1.4(1.4) | 20.4(15.8) |
| CQC(n=56) | 73.5(30.5) | 17.2(8.2) | 4.8(1.6) | 6.5(6.2) | 19.7(9.6) | 6.1(2.7) | 7.4(3.5) | 0.6(0.4) | 11.2(6.1) |
| Background sites | | | | | | | | | |
| XLZ(n=42) | 42.6(20.1) | 12.4(5.1) | 1.5(0.7) | 3.7(5.0) | 8.4(7.0) | 3.4(2.2) | 5.0(2.7) | 0.3(0.3) | 7.9(5.6) |
| LAZ(n=60) | 66.3(36.6) | 21.7(6.5) | 2.9(1.4) | 8.7(8.5) | 11.2(6.3) | 7.3(4.5) | 2.0(2.0) | 0.6(0.8) | 11.9(8.2) |
| DHM(n=36) | 40.1(20.4) | 11.6(5.0) | 2.0(1.0) | 4.5(3.9) | 10.1(5.3) | 4.0(1.7) | 3.8(0.9) | 0.5(0.6) | 3.6(1.5) |
| NMT(n=35) | 9.5(10.7) | 3.4(2.7) | 0.2(0.5) | 0.1(0.1) | 0.4(0.4) | 0.4(0.2) | 3.9(2.0) | 0.1(0.0) | 1.1(2.6) |
| CBM(n=52) | 23.3(6.8) | 8.9(3.6) | 0.9(0.6) | 1.1(1.4) | 3.3(2.3) | 1.8(0.9) | 3.7(1.9) | 0.2(0.2) | 3.5(3.4) |
| GGM(n=36) | 32.2(29.7) | 13.1(13.5) | 1.1(0.8) | 0.4(0.5) | 4.7(4.1) | 1.7(1.3) | 3.2(2.9) | 0.4(1.4) | 7.7(8.0) |

 $^*$MD: mineral dust; $^{**}$Unaccounted: the difference between the $PM_{2.5}$ gravimetric mass and the sum of the PM
 constituents (OM, EC, $SO_4^{2-}$, $NO_3^-$, $NH_4^+$, Mineral dust and $Cl^-$).


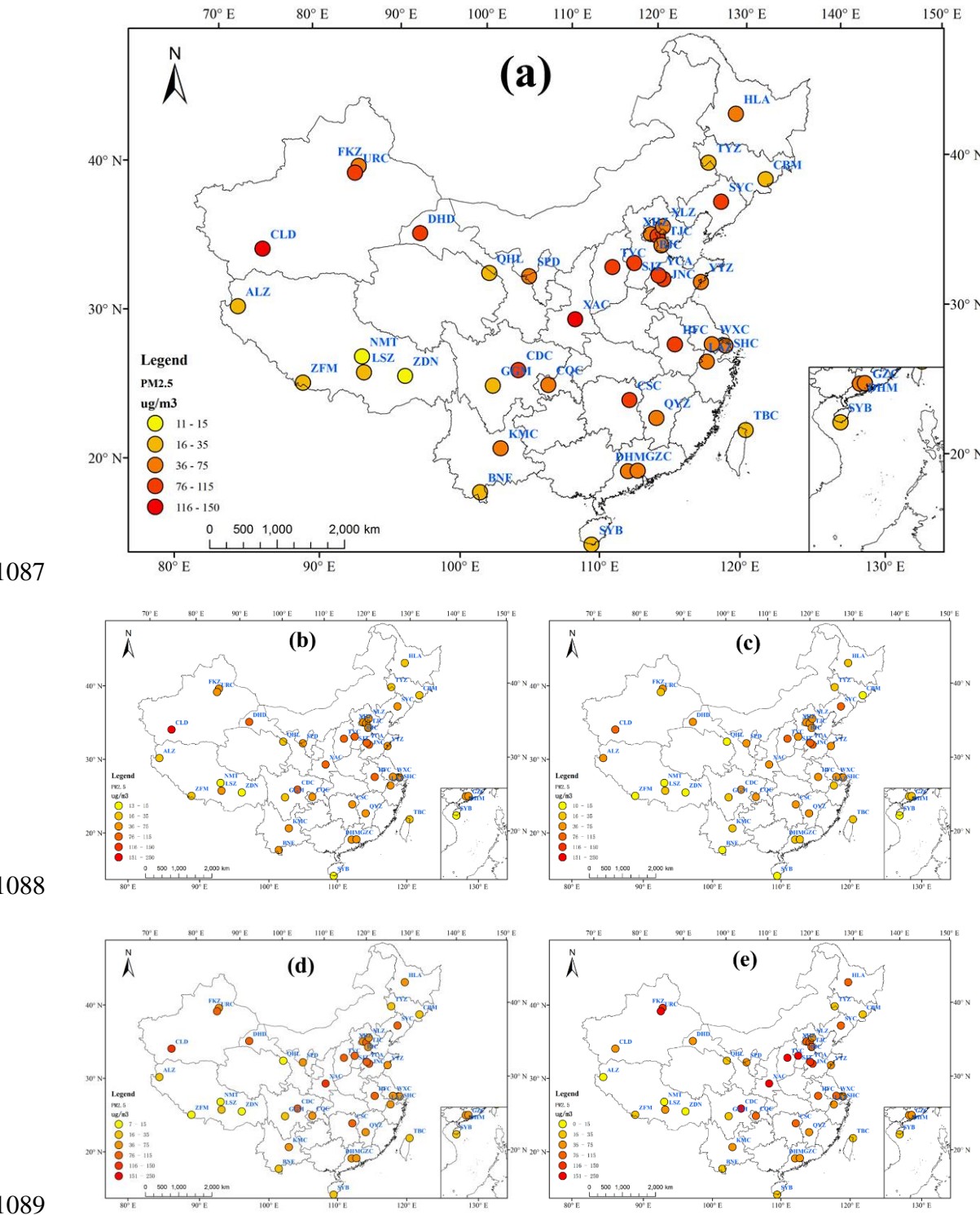



Fig.1. Locations and the averaged PM$_{2.5}$ concentrations of the forty monitor stations during (a) the year of 2012-2014, (b) spring, (c) summer, (d) autumn and (e) winter. The site code related to the observation stations could be found in Table 1.


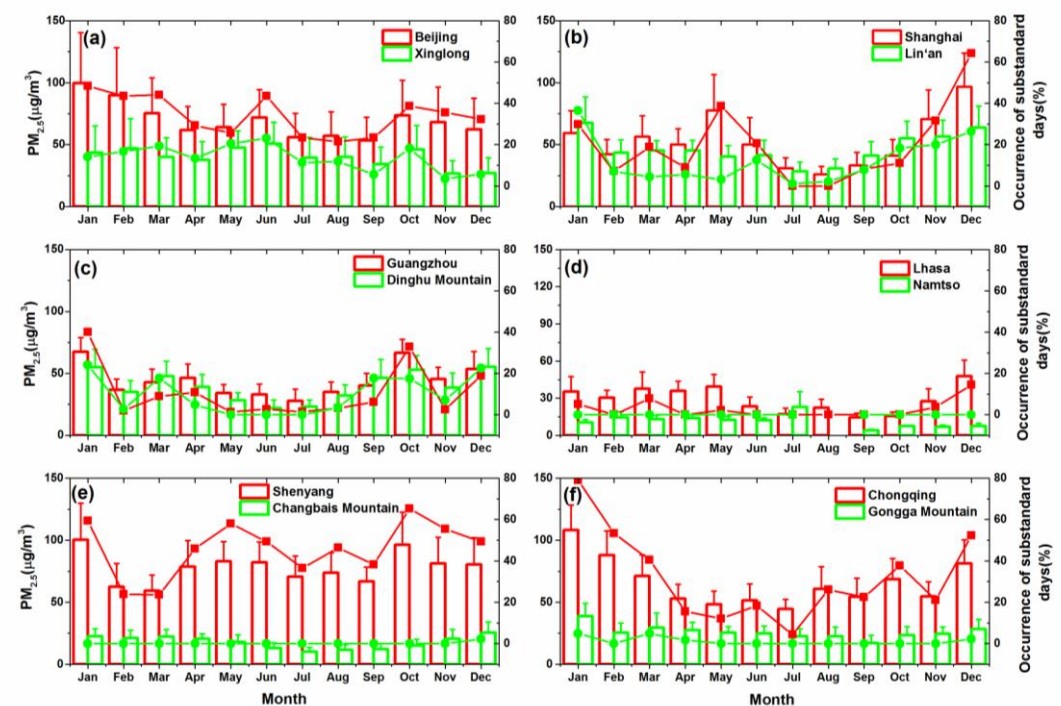

Fig.2. Monthly average PM$_{2.5}$ concentration (histogram, left coordinate) and the occurrence of substandard days in each month (dotted line, right coordinate) at urban and background sites in (a)North China plain, (b)Yangtze River delta, (c) Pearl River delta, (d)Tibetan Autonomous Region , (e) Northeast China Region and (f) Southwestern China Region. The error bars stands for the standard deviation.

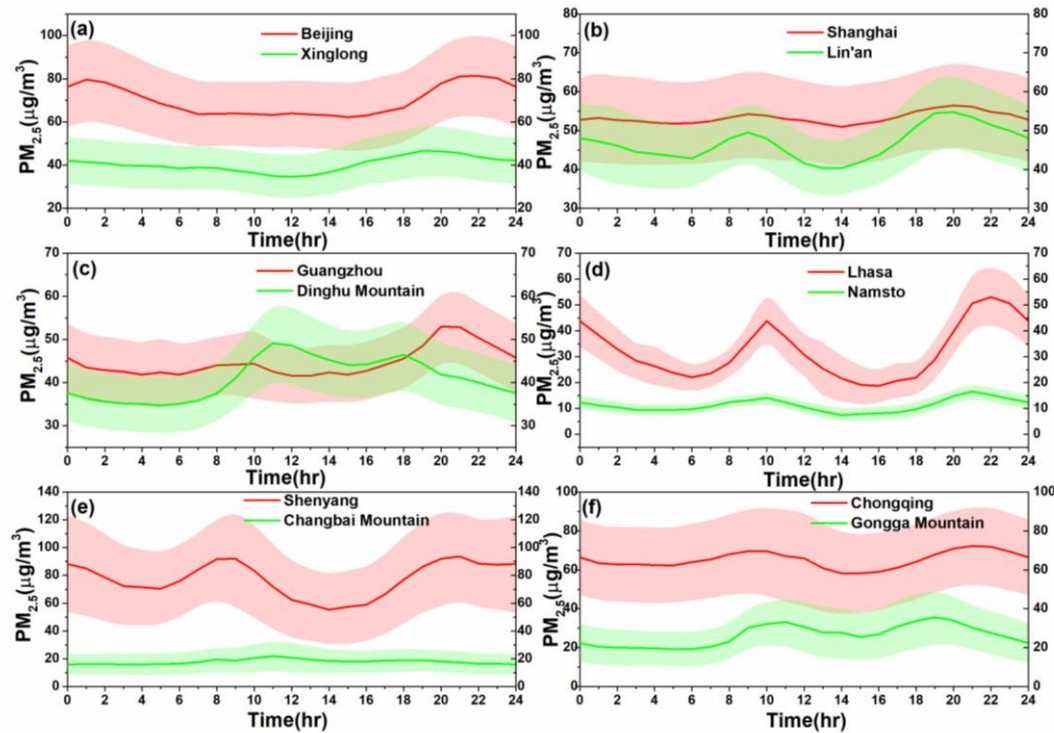


Fig.3 Diurnal cycles of PM$_{2.5}$ at six paired urban and background sites in (a)North China plain,
(b)Yangtze River delta, (c) Pearl River delta, (d)Tibetan Autonomous Region, (e) Northeast China
Region and (f) Southwestern China Region. Shadow area represent the error bars and stands for one
half of the standard deviation.

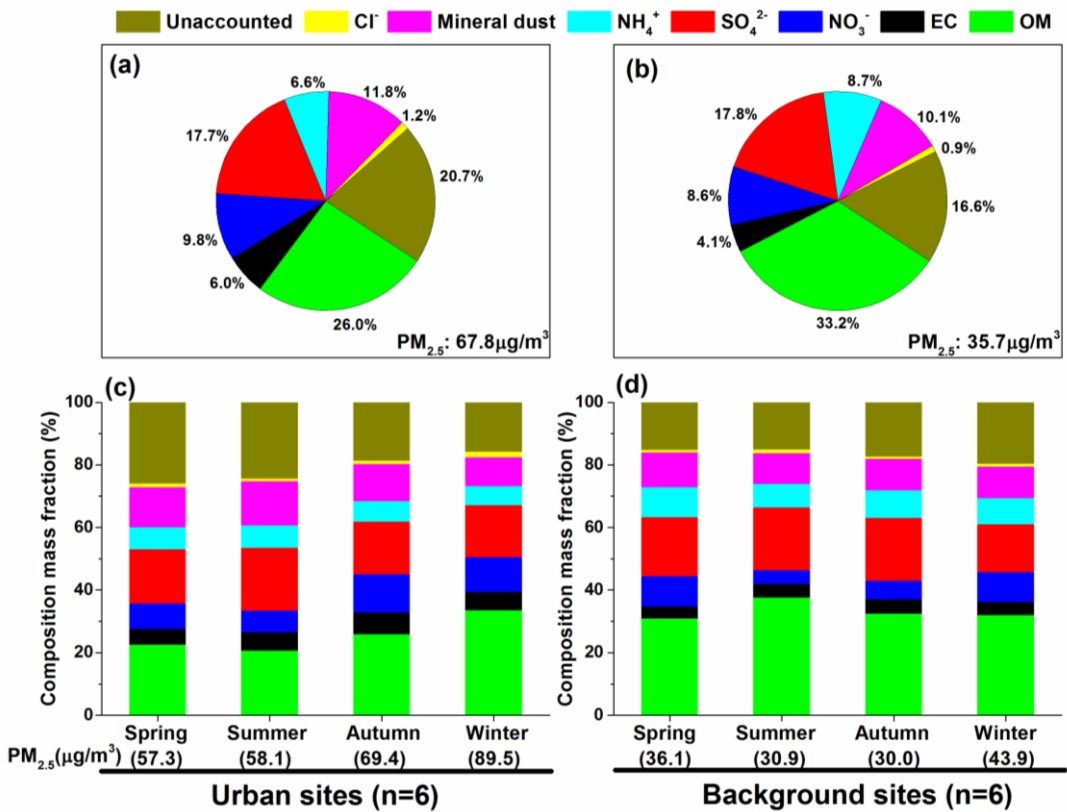

Fig. 4 Average chemical composition and its seasonal variations of PM$_{2.5}$ in (a, c) urban sites and (b, d) background sites. The unaccounted matter refer to the difference between the PM$_{2.5}$ gravimetric mass and the sum of the PM constituents (OM, EC, SO$_4^{2-}$, NO$_3^-$, NH$_4^+$, Mineral dust and Cl$^-$).

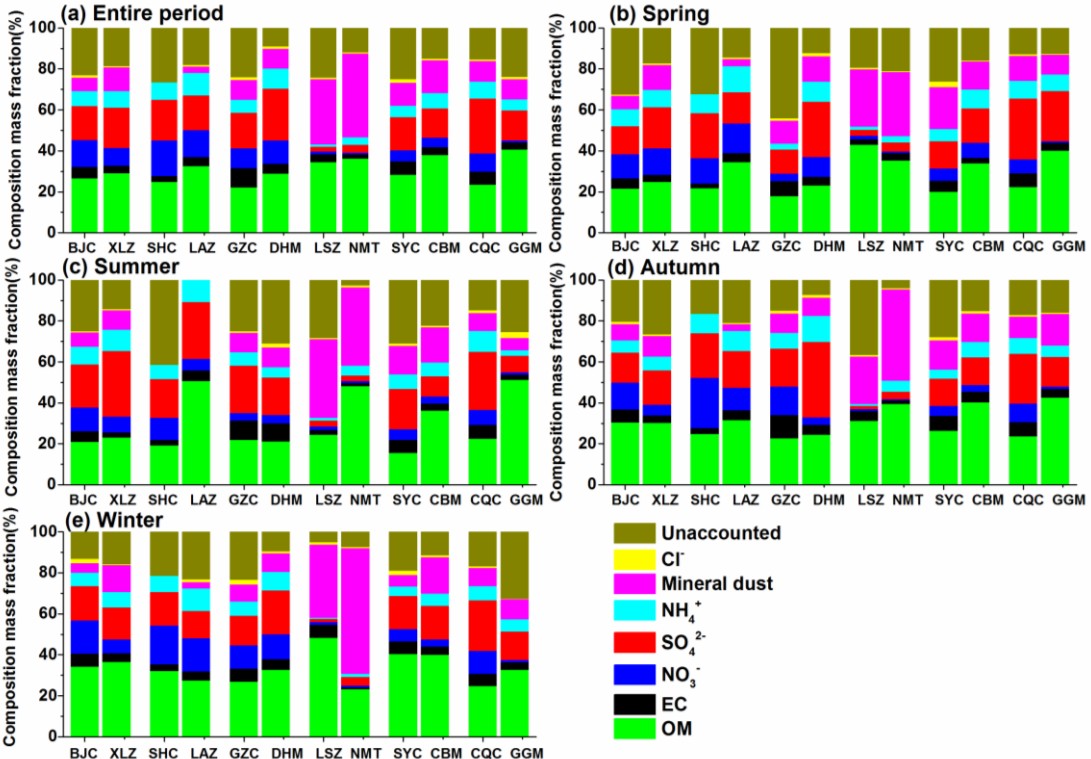

Fig.5 Average chemical composition of PM$_{2.5}$ in individual site during (a) the entire period and (b-
e) the different seasons. The unaccounted matter refer to the difference between the PM$_{2.5}$
gravimetric mass and the sum of the PM constituents (OM, EC, SO$_4^{2-}$, NO$_3^-$, NH$_4^+$,Mineral dust and
Cl$^-$). The site code related to the observation stations could be found in Table 1.

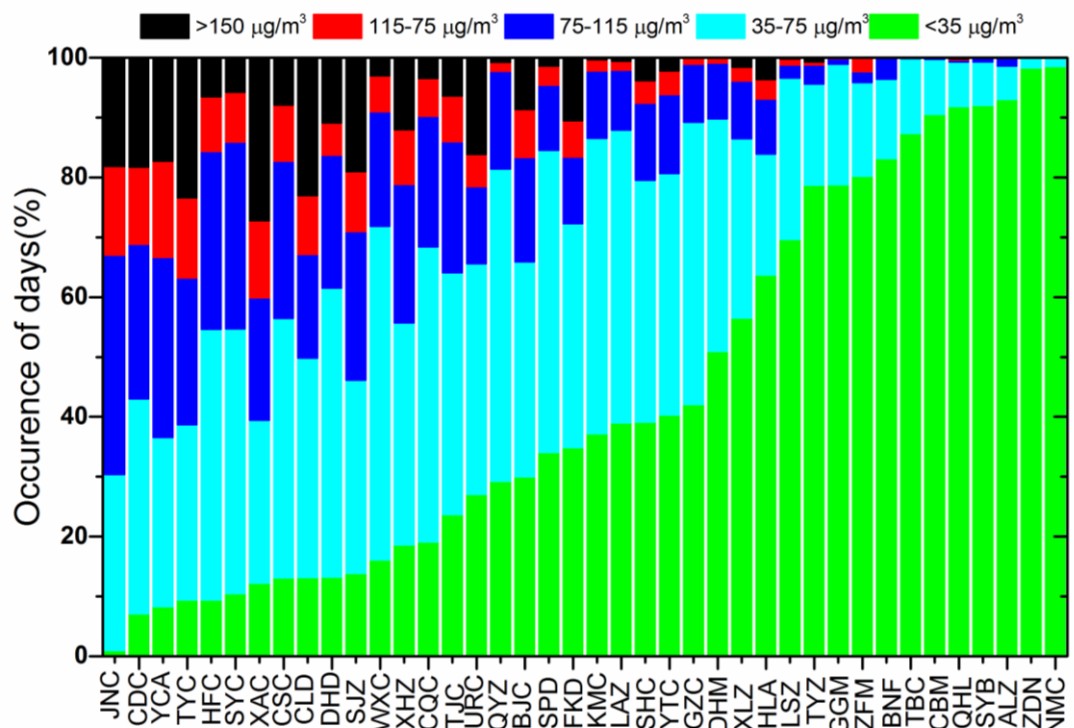


Fig.6 Days separated by the threshold values of the "Ambient Air Quality Standard" (AAQS)
(GB3095-2012) of China guideline. The threshold values of 35, 75, 115 and 150μg/m³ used for the
daily concentration ranges are represented as clean (<35μg/m³), slightly polluted (35-75μg/m³),
moderated polluted (75-115μg/m³), polluted (115-150μg/m³) and heavily polluted (>150μg/m³),
which suggested by the guideline of the AAQS. The site code related to the observation stations
could be found in Table 1.

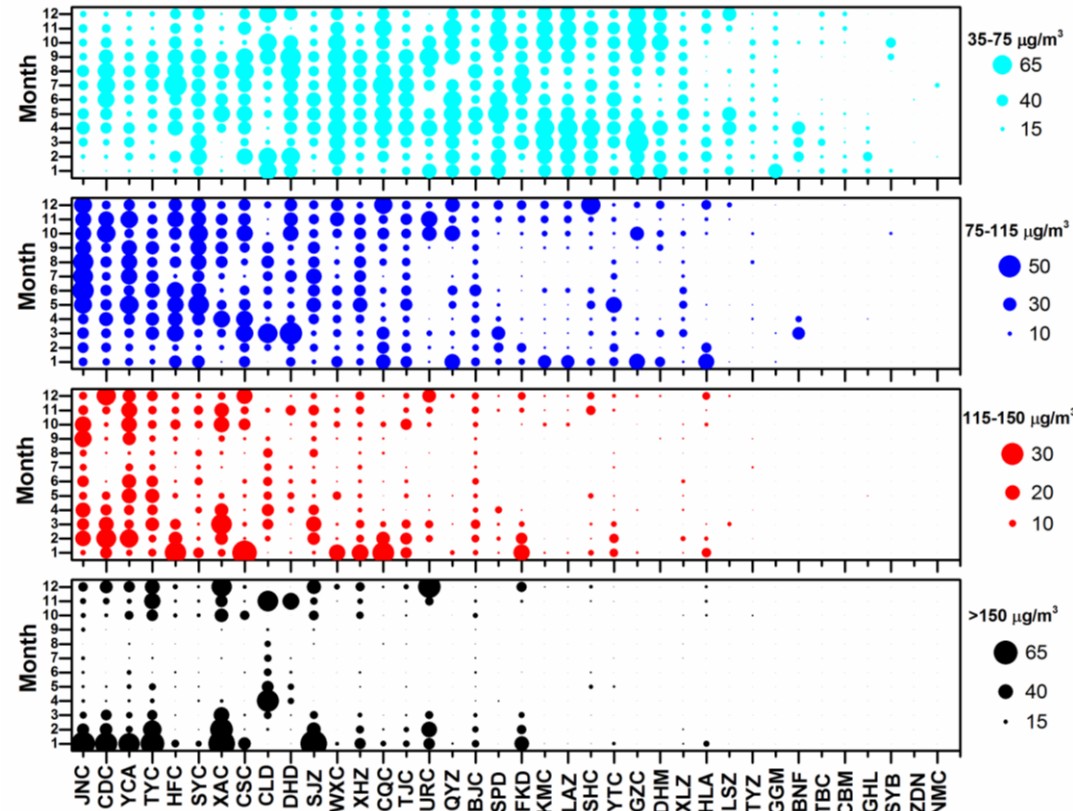

Fig.7 Monthly distribution of the occurrence of the polluted days exceeding the "Ambient Air
Quality Standard" (AAQS) (GB3095-2012) of China. The symbol size represents the occurrences
of polluted days for the corresponding month. The symbol color represents the different mass range.
The sites of Nagri and Mount Everest are excluded because of the small sample size. The site code
related to the observation stations could be found in Table 1.


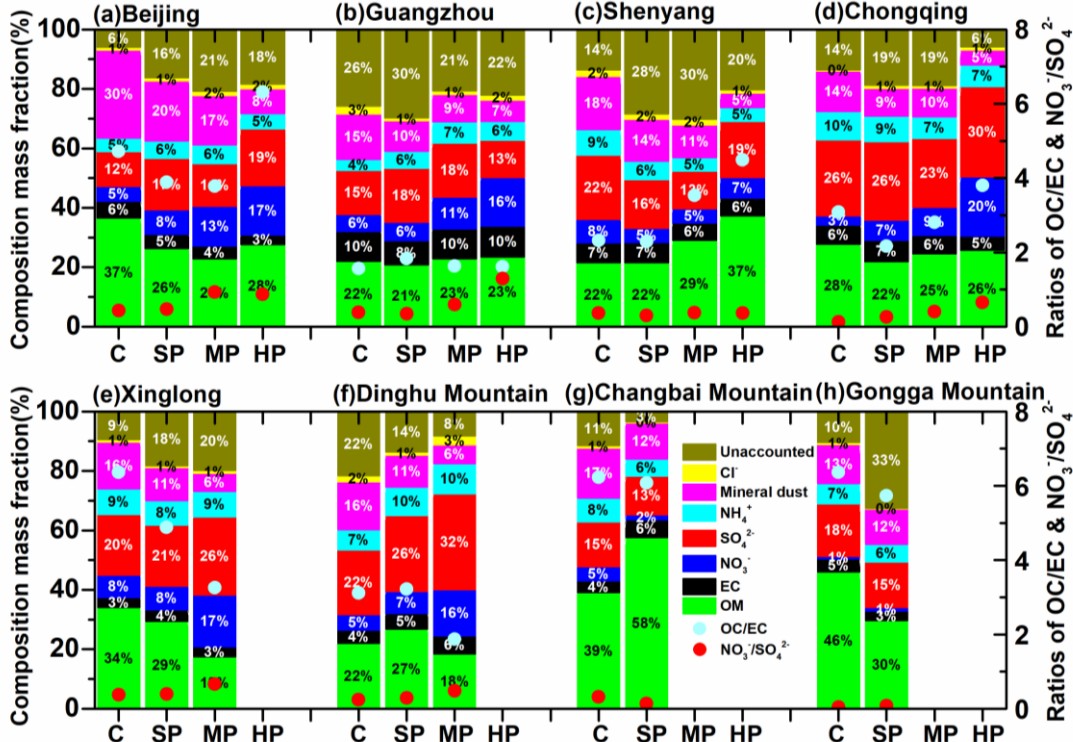

Fig. 8 Average chemical composition of $PM_{2.5}$ and the mass ratio of $[NO_3^-]/[SO_4^{2-}]$ and OC/EC with
respect to pollution level. The C, SP, MP and HP is related to clean (daily $PM_{2.5}$ <35 μg/m³), slightly
polluted (35 μg/m³<daily $PM_{2.5}$ <75 μg/m³), moderated polluted (75 μg/m³<daily $PM_{2.5}$ <150 μg/m³)
and heavily polluted (daily $PM_{2.5}$ >150 μg/m³). The unaccounted matter refer to the difference
between the $PM_{2.5}$ gravimetric mass and the sum of the PM constituents (OM, EC, $SO_4^{2-}$, $NO_3^-$,
$NH_4^+$, Mineral dust and $Cl^-$).