# Peer review of "Characteristics of PM2.5 mass concentrations and chemical species in urban and background"

_Atmospheric Chemistry and Physics, 2017_

## Referee Comment (RC1) · Anonymous Referee #1 · 21 Feb 2018

The paper presents the first long-term datasets from the "Campaign on atmospheric Aerosol REsearch" network of China (CARE-China), including three years of observations of online PM2.5 mass concentrations (2012-2014) and one year of observations of PM2.5 compositions (2012-2013). The average PM2.5 concentrations at 20 urban sites was three times higher than the average value from the 12 background sites. The PM2.5 concentrations are generally higher in east-central China than in the other parts of the country due to their relative large particulate matter (PM) emissions and the unfavourable meteorological conditions for pollution dispersion. The seasonal variabil-
ity of the PM2.5 shows high values in winter and low values during summer at urban sites. Bimodal and unimodal diurnal variation patterns were identified at both urban and background sites. The chemical compositions of PM2.5 at all urban sites are organic matter (OM), $SO_4^{2-}$, mineral dust, $NO_3^-$, $NH_4^+$, elemental carbon (EC), $Cl^-$ at 45% RH and residual matter (20.7%). Similar chemical compositions of PM2.5 were observed at background sites but were associated with higher fractions of OM and lower fractions of $NO_3^-$ and EC. Significant variations of the chemical species were observed among the sites. The PM2.5 chemical species at the background sites exhibited larger spatial heterogeneities than those at urban sites. Six pairs of urban and background sites from each region of China were selected, and the differences in the chemical compositions of urban and background sites were analysed. It is suggested that there are different contributions from regional anthropogenic or natural emissions and from the long-range transport to background areas. Notable seasonal variations of PM2.5 polluted days were observed, especially for the megacities in east-central China, resulting in frequent heavy pollution episodes occurring during winter. General comments It is concluded from the similar evolution of the PM2.5 chemical compositions on polluted days at the urban and nearby background sites that there are significant regional pollution characteristics of the most polluted areas of China. Following this it is stated that the chemical species dominating the evolutions of the heavily polluted events were different in these areas, indicating that unique mitigation measures should be developed for different regions of China. This is not conclusive and must be explained in more detail: What means "significant regional pollution characteristics of the most polluted areas of China" together with "chemical species dominating the evolutions of the heavily polluted events were different in these areas"? What means "unique mitigation measures should be developed for different regions of China"? This more precise description is required due to the conclusion that the analyses provides insights into the sources, processes, and lifetimes of heavily polluted events. The paper addresses relevant scientific tasks. The paper presents novel concepts, ideas and tools. The scientific methods and assumptions are valid and clearly outlined so that

substantial conclusions are reached. The description of experiments and calculations allow their reproduction by fellow scientists. The quality of the figures is good. The figure captions should be improved so that these are understandable without the overall manuscript. The related work is well cited so that the authors give proper credit to related work and own new contribution. The title as well as the abstract reflects the whole content of the paper. The overall presentation is well structured and clear. The language is fluent. The mathematical formulae, symbols, abbreviations, and units are generally correctly defined and used. Specific Comments Different instruments for measurements of PM2.5 mass concentrations are applied at the different sites and well described. But what shows an intercomparison of these different types of instruments? Chapter 4 is a summary with some conclusions. More detailed conclusions are possible and should be drawn. Technical corrections Unaccounted and residual matter is for the same in chemical composition. This should be explained – what does it mean? Some free spaces are missing in the figure captions.

---

## Referee Comment (RC2) · Anonymous Referee #2 · 30 Mar 2018

General comment:

This paper presents three-year dataset of mass concentrations and chemical composition of PM2.5 at multiple urban and background sites in China. The chemical composition quantified includes organic matter, elemental carbon, sulfate, nitrate, ammonium, mineral dusts, and chlorine. Such spatial and temporal data are valuable addition to the literature. The manuscript is well written and organized. Therefore, I suggest publication with minor revision.

[Figure]

Specific comments 1. P2, Line 49-52, explain with 1-2 sentences what the difference during the evolutions of the heavily polluted events. 2. P4, line 145-160, two types of instruments (TEOM and EBAM) were used for measurements of PM2.5 mass concentrations at the different sites. The authors should provide inter-comparison results for better quality control. 3. P5, line 174, "The sampling lasted 24 or 48 h", please clarify the sampling scheme. 4. P5, line 179-190, the information about the chemical analysis is insufficient, more details about the calibration, performance such as detection limits should be provided. 5. P6-7, line 249-279, better to combine the last two paragraphs of section 3.1.1 and discuss the comparison between this study and those from the Europe, North America and the other parts of world by grouping the data into urban, rural, and background sites. Also, better to compare with literature data measured at the same sites and periods. 6. P9, line 350-352, the invisible morning peak of PM2.5 in Beijing, Shanghai and Guangzhou is interesting, which was somewhat different from the previous studies. Not clear, please explain. 7. P9, line 360-361. The discussion of bimodal pattern of PM2.5 in Lin'an is not convincing, more likely Lin'an is highly affected by the regional transportation from the YRD region, which was totally different from Namsto and Gongga Mountain. 8. P10, line 392-394, need reference here for the calculate method of Si. 9. P11, line 432-437, the explanation is not convincing, the higher fraction of sulfate in south China is more likely associated to the higher oxidation capacity in south China and therefore higher formation efficiency from SO2 to SO42-. 10. P11, line 456-460, what about the fraction of EC? Is it higher or lower compared with previous studies? More discussion about the comparisons with previous studies would be useful, at least for the urban sites. 11. P16, better to split section 3.3 into two sub-sections. One section focuses on the mass concentration of PM2.5 and the other on the chemical composition of PM2.5. 12. P17, line 722-724, "These results suggest the different formation mechanisms of the heavy pollution in the most polluted city clusters, and unique mitigation measures should be developed for the different regions of China." This conclusion is ambiguous, the authors should clearly state the difference in formation mechanisms, and the implications for mitigation measures. 13.

P26, Table 2, please provide the standard deviation of the concentration of PM2.5 and its components. 14. P31, Fig.5, need a clear figure caption.

**[ACPD](https://...)**

---

## Author Comment (AC1) · 28 May 2018

**Response to Anonymous Referee #1**

We appreciate your valuable comments and suggestion, which significantly improved the manuscript. We carefully answered them point-by-point as below and improved the corresponding parts in the manuscript.
Reviewer's comments are in plain face.
Author's responses are in blue color.
Changes in the manuscript are in red color.

The paper presents the first long-term datasets from the "Campaign on atmospheric Aerosol REsearch" network of China (CARE-China), including three years of observations of online $PM_{2.5}$ mass concentrations (2012-2014) and one year of observations of $PM_{2.5}$ compositions (2012-2013). The average $PM_{2.5}$ concentrations at 20 urban sites was three times higher than the average value from the 12 background sites. The $PM_{2.5}$ concentrations are generally higher in east-central China than in the other parts of the country due to their relative large particulate matter (PM) emissions and the unfavourable meteorological conditions for pollution dispersion. The seasonal variability of the $PM_{2.5}$ shows high values in winter and low values during summer at urban sites. Bimodal and unimodal diurnal variation patterns were identified at both urban and background sites. The chemical compositions of $PM_{2.5}$ at all urban sites are organic matter (OM), $SO_4^{2-}$, mineral dust, $NO_3^-$, $NH_4^+$, elemental carbon (EC), $Cl^-$ at 45% RH and residual matter (20.7%). Similar chemical compositions of $PM_{2.5}$ were observed at background sites but were associated with higher fractions of OM and lower fractions of $NO_3$- and EC. Significant variations of the chemical species were observed among the sites. The $PM_{2.5}$ chemical species at the background sites exhibited larger spatial heterogeneities than those at urban sites. Six pairs of urban and background sites from each region of China were selected, and the differences in the chemical compositions of urban and background sites were analysed. It is suggested that there are different contributions from regional anthropogenic or natural emissions and from the long-range transport to background areas. Notable seasonal variations of $PM_{2.5}$ polluted days were observed, especially for the megacities in east-central China, resulting in frequent heavy pollution episodes occurring during winter.
[Response] Thank you very much for your comments.

**General comments**

It is concluded from the similar evolution of the $PM_{2.5}$ chemical compositions on polluted days at the urban and nearby background sites that there are significant regional pollution characteristics of the most polluted areas of China. Following this it is stated that the chemical species dominating the evolutions of the heavily polluted events were different in these areas, indicating that unique mitigation measures should be developed for different regions of China. This is not conclusive and must be explained in more detail: What means "significant regional pollution characteristics of the most polluted areas of China" together with "chemical species dominating the evolutions of the heavily polluted events were different in these areas"? What means "unique mitigation measures should be developed for different regions of China"? This more precise description is required due to the conclusion that the analyses provides insights into the sources, processes, and lifetimes of heavily polluted events.

The paper addresses relevant scientific tasks. The paper presents novel concepts, ideas and tools. The scientific methods and assumptions are valid and clearly outlined so that substantial conclusions are reached. The description of experiments and calculations allow their reproduction by fellow scientists. The quality of the figures is good. The figure captions should be improved so that these are understandable without the overall manuscript.

The related work is well cited so that the authors give proper credit to related work and own new contribution. The title as well as the abstract reflects the whole content of the paper. The overall presentation is well structured and clear. The language is fluent. The mathematical formulae, symbols, abbreviations, and units are generally correctly defined and used.

[Response] Thank you very much for your comments, and thanks for the affirmation of reviewer to our work.

(1) The "significant regional pollution characteristics of the most polluted areas of China" means fine particle pollution in the most polluted areas of China assumes a regional tendency, according to the consistent evolution of fine particle chemical composition between urban site and its nearby background site. Sorry for the misunderstanding. In addition, we admitted that "chemical species dominating the evolutions of the heavily polluted events were different in these areas" is ambiguous. To make it clear, we revised these sentences in the Abstract as showed below:

"The evolution of the $PM_{2.5}$ chemical compositions on polluted days was consistent for the urban and nearby background sites, where the sum of sulfate, nitrate and ammonia typically constituted much higher fractions (31-57%) of $PM_{2.5}$ mass, suggesting fine particle pollution in the most polluted areas of China assumes a regional tendency, and the importance to address the emission reduction of secondary aerosol precursors including $SO_2$ and NOx."

(2) Sorry for the misunderstanding. We admitted that "unique mitigation measures should be developed for different regions of China" is ambiguous. To make it clear, more discussion about the major primary sources contributed to the high fine particle loading in specific regions was conducted in section 3.3.2. Based on these analysis, we revised the sentences as follow:

"Furthermore, distinct differences in the evolution of $[NO_3^-]/[SO_4^{2-}]$ ratio and OC/EC ratio in polluted days imply that mobile sources and stationary (coal combustion) sources are likely more important in Guangzhou and Shenyang, respectively, whereas in Beijing it is mobile sources and biomass burning. As for Chongqing, the higher oxidation capacity than the other three cities suggested it should pay more attention to the emission reduction of secondary aerosol precursors."

(3) The figure captions have been improved.

**Specific Comments:**

Different instruments for measurements of $PM_{2.5}$ mass concentrations are applied at the different sites and well described. But what shows an intercomparison of these different types of instruments?

[Response] Thank you for your comments. In fact, the intercomparison of the different types of instruments had been done before the routine work of CARE-China network. First, we provide more details on the comparison of $PM_{2.5}$ mass concentration measured from two kinds of on-line instruments (TEOM and EBAM) used in this study, the results showed that these two on-line instruments correlated well ($R^2=0.90$, P<0.001). TEOM reported approximately 24% lower mass concentration than EBAM, and the difference could be explained by the loss of semi-volatile materials from TEOM (Zhu et al., 2007).

Second, the comparison of $PM_{2.5}$ mass concentration measured from filter sampling and the on-line instruments (TEOM and EBAM) during the one-year observation period was provided. On average, $PM_{2.5}$ mass concentrations measured by the filter sampling was approximately 9% higher than the on-line instruments. The discussions about the intercomparison was added in section 2.2 and the results was provided in the support information.

[Figure]

Fig. S1 (a) Intercomparison of $PM_{2.5}$ mass concentrations measured by the tapered element oscillating microbalance (TEOM) and the beta gauge instruments (EBAM) conducted at the Beijing site; (b) Intercomparison of $PM_{2.5}$ mass concentrations measured by filter sampling and the on-line instruments (TOEM and EBAM) from the 11 sites during the one-year observation period. (BJC: Beijing; XLZ: Xinglong; SYC: Shenyang; CBM: Changbai Mountain; LSZ: Lhasa; NMT: Namtso; CQZ: Chongqing; GGM: Gongga Mountain; GZC: Guangzhou; DHM: Dinghu Mountain; LAZ: Lin'an)

Zhu, K., Zhang, J., Lioy, P. J.: Evaluation and Comparison of Continuous Fine Particulate Matter Monitors for Measurement of Ambient Aerosols, J. Air & Waste Manage. Assoc., 57:12, 1499-1506, 2007.

Chapter 4 is a summary with some conclusions. More detailed conclusions are possible and should be drawn.

[Response] Thank you for your comments. We revised Chapter 4, and more detailed conclusions from the chemical evolution of $PM_{2.5}$ composition in polluted days and the implication for the mitigation measures were drawn in the revised MS.

"…The increasing contribution of secondary aerosol on polluted days was observed both for the urban and nearby background sites, suggesting fine particle pollution in the most polluted areas of China assumes a regional tendency, and the importance to address the emission reduction of secondary aerosol precursors. In addition, the chemical species dominating the evolutions of the heavily polluted events were different, while decreasing or constantly contribution of OM associated with increasing contribution of SIA characteristic evolution of $PM_{2.5}$ in NCP, PRD and SWCR, the opposite phenomenon was observed in NECR. Further analysis from the $[NO_3^-]/[SO_4^{2-}]$ ratio and OC/EC ratio showed that fine particle pollution in Guangzhou and Shenyang was mainly attributed to the traffic emissions and coal combustion, respectively, while more complex and variable major sources including mobile vehicle emission and residential sources contributed to the development of heavily polluted days in Beijing. As for Chongqing, the higher oxidation capacity than other cities suggested it should pay more attention to the emission reduction of secondary aerosol precursors. These results suggest the different formation mechanisms of the heavy pollution in the most polluted city clusters, and unique mitigation measures should be developed for the different regions of China."

**Technical corrections:**

Unaccounted and residual matter is for the same in chemical composition. This should be explained – what does it mean? Some free spaces are missing in the figure captions.

[Response] Thank you for your comments. Yes, the unaccounted and residual matter are the same which both refer to the difference between the $PM_{2.5}$ gravimetric mass and the sum of the PM constituents (OM, EC, $SO_4^{2-}$, $NO_3^-$, $NH_4^+$, Mineral dust and $Cl^-$). The remaining unaccounted-for mass fraction may be the result of analytical errors, a systematic underestimation of the PM constituents whose concentrations are calculated from the measured data (e.g., OM, and mineral dust), and aerosol-bound water (especially when mass concentrations are determined at RH >30%). To make it clear, "residual matter" was replaced by "unaccounted" throughout the MS, for consistency. In addition, the figure captions have been improved.

[revised manuscript text omitted]

---

## Author Comment (AC2) · 28 May 2018

**Response to Anonymous Referee #2**

We appreciate your valuable comments and suggestion, which significantly improved the manuscript. We carefully answered them point-by-point as below and improved the corresponding parts in the manuscript.

Reviewer's comments are in plain face.

Author's responses are in blue color.

Changes in the manuscript are in red color.

**General comment:**

This paper presents three-year dataset of mass concentrations and chemical composition of $PM_{2.5}$ at multiple urban and background sites in China. The chemical composition quantified includes organic matter, elemental carbon, sulfate, nitrate, ammonium, mineral dusts, and chlorine. Such spatial and temporal data are valuable addition to the literature. The manuscript is well written and organized. Therefore, I suggest publication with minor revision.

[Response] Thank you very much for your comments, and thanks for the affirmation of reviewer to our work.

Specific comments

1.  P2, Line 49-52, explain with 1-2 sentences what the difference during the evolutions of the heavily polluted events.

(1) [Response] Thanks for the suggestion. To make it clear, more discussion about the major primary sources contributed to the high fine particle loading in the specific region was conducted in section 3.3.2. Based on these analysis, we revised the sentences as follow:

"Furthermore, distinct differences in the evolution of $[NO_3^-]/[SO_4^{2-}]$ ratio and OC/EC ratio in polluted days imply that mobile sources and stationary (coal combustion) sources are likely more important in Guangzhou and Shenyang, respectively, whereas in Beijing it is mobile sources and residential emissions. As for Chongqing, the higher oxidation capacity than the other three cities suggested it should pay more attention to the emission reduction of secondary aerosol precursors."

2.  P4, line 145-160, two types of instruments (TEOM and EBAM) were used for measurements of $PM_{2.5}$ mass concentrations at the different sites. The authors should provide inter-comparison results for better quality control.

[Response] We agree with the reviewer's comments. In fact, the intercomparison of the different types of instruments had been done before the routine work of CARE-China network. Details on the comparison of $PM_{2.5}$ mass concentration measured from two kinds of on-line instruments (TEOM and EBAM) have been provided in the Methods section and also in the support information.

"A year-long intercomparison of daily $PM_{2.5}$ mass concentrations measured by TEOM and EBAM was conducted at the Beijing site (Fig. S1a), and the results showed that these two on-line instruments correlated well ($R^2$=0.90, P<0.01). TEOM reported approximately 24% lower mass concentration than EBAM, and the difference could be explained by the loss of semi-volatile materials from TEOM (Zhu et al., 2007). "

[Figure]

Fig. S1 (a) Intercomparison of PM$_{2.5}$ mass concentrations measured by the tapered element oscillating microbalance (TEOM) and the beta gauge instruments (EBAM) conducted at the Beijing site; (b) Intercomparison of PM$_{2.5}$ mass concentrations measured by filter sampling and the on-line instruments (TOEM and EBAM) from the 11 sites during the one-year observation period. (BJC: Beijing; XLZ: Xinglong; SYC: Shenyang; CBM: Changbai Mountain; LSZ: Lhasa; NMT: Namtso; CQZ: Chongqing; GGM: Gongga Mountain; GZC: Guangzhou; DHM: Dinghu Mountain; LAZ: Lin'an)

Zhu, K., Zhang, J., Lioy, P. J.: Evaluation and Comparison of Continuous Fine Particulate Matter Monitors for Measurement of Ambient Aerosols, J. Air & Waste Manage. Assoc., 57:12, 1499-1506, 2007.

3. P5, line 174, "The sampling lasted 24 or 48 h", please clarify the sampling scheme.
[Response] Sorry for the misunderstanding. The sampling scheme at Guangzhou site has been clarified.
"The sampling lasted 48h for the first three samples and 24 h for the rest samples, generally starting at 8:00 a.m."

4. P5, line 179-190, the information about the chemical analysis is insufficient, more details about the calibration, performance such as detection limits should be provided.
[Response] Thanks for the suggestion. More details about the calibration, performance such as detection limits have been provided in the revised MS.
"Three types of chemical species were measured using the methods described in Xin et al. (2015). Briefly, the organic carbon (OC) and elemental carbon (EC) values were determined using a thermal/optical reflectance protocol using a DRI model 2001 carbon analyzer (Atmoslytic, Inc., Calabasas, CA, USA) with the thermal/optical reflectance (TOR) method. A circle piece of 0.495 cm$^2$ was cut off from the filters and was sent into the thermal optical carbon analyzer. In a pure helium atmosphere, OC1, OC2, OC3 and OC4 are produced stepwise at 140 °C, 280 °C, 480 °C and 580 °C, respectively; followed by EC1 (540 °C), EC2 (780 °C) and EC3 (840 °C) in a 2% oxygen-contained helium atmosphere. Eight main ions, including K$^+$, Ca$^{2+}$, Na$^+$, Mg$^{2+}$, NH$_4^+$, SO$_4^{2-}$, NO$_3^-$

and Cl⁻, were measured via ion chromatography (using a Dionex DX 120 connected to a DX AS50 autosampler for anions and a DX ICS90 connected to a DX AS40 autosampler for cations). One-quarter of each filter substrate was extracted with 25 mL deionized water in a PET vial for 30 min. Before performing a targeted sample analysis, a standard solution and blank test were performed, and the correlation coefficient of the standard samples was more than 0.999. The detection limits for all anions and cations, which were calculated as three times the standard deviations of seven replicate blank samples, are all lower than 0.3 μg m⁻³. The microwave acid digestion method was used to digest the filter samples into liquid solution for elemental analysis. One quarter of each filter sample was placed in the digestion vessel with a mixture of 6 mL $HNO_3$, 2 mL $H_2O_2$ and 0.6 mL HF, and was then exposed to a three-stage microwave digestion procedure from a microwave-accelerated reaction system (MARS, CEM Corporation, USA). After that, 18 elements, including Mg, Al, K, Ca, V, Cr, Mn, Fe, Co, Ni, Cu, Zn, As, Se, Ag, Cd, Tl and Pb, were determined by Agilent 7500a inductively coupled plasma mass spectrometry (ICP-MS, Agilent Technologies, Tokyo, Japan). Quantification was carried out by the external calibration technique using a set of external calibration standards (Agilent Corporation) at concentration levels close to that of the samples. The relative standard deviation for each measurement (repeated twice) was within 3%. The method detection limits (MDLs) were determined by adding 3 standard deviations of the blank readings to the average blank values (Yang et al., 2009). Quality control and quality assurance procedures were routinely applied for all the carbonaceous, ion and elemental analysis. ”

5. P6-7, line 249-279, better to combine the last two paragraphs of section 3.1.1 and discuss the comparison between this study and those from the Europe, North America and the other parts of world by grouping the data into urban, rural, and background sites. Also, better to compare with literature data measured at the same sites and periods.
[Response] Thanks for the suggestion. We have rewrite this part followed your suggestions.

“For urban/suburban sites, average PM$_{2.5}$ concentrations of 20.1 μg/m³ was reported by Gehrig and Buchmann (2003) from 1998 to 2001 in Switzerland, and average concentrations of 16.3 μg/m³ for the period 2008-2009 in the Netherlands (Janssen et al., 2013). Between October 2008 and April 2011, the 20 study areas covered major cities of the European ESCAPE project showed annual average concentrations of PM$_{2.5}$ ranging from 8.5 to 29.3 μg/m³, with low concentrations in northern Europe and high concentrations in southern and eastern Europe (Eeftens et al., 2012). Constructed a database of PM$_{2.5}$ component concentrations from 187 counties in the United States for 2000-2005, Bell et al. (2007) reported an average PM$_{2.5}$ value of 14.0 μg/m³, with higher values in the eastern United States and California, and lowest values in the central regions and Northwest. For background sites, Putaud et al. (2010) showed that annual average of PM$_{2.5}$ ranged from 3 to 22μg/m³ observed from 12 background sites across Europe. In addition, average PM$_{2.5}$ value of 12.6μg/m³ was observed at a regional background site in the Western Mediterranean from 2002 to 2010 (Cusack et al., 2012).”

6. P9, line 350-352, the invisible morning peak of PM$_{2.5}$ in Beijing, Shanghai and Guangzhou is interesting, which was somewhat different from the previous studies. Not clear, please explain.
[Response] Thanks for pointing out this. Yes, the invisible morning peak of PM$_{2.5}$ in

Beijing, Shanghai and Guangzhou is different from those studies based on the historical records of PM$_{2.5}$ (Zhao, et al., 2009), but is similar with those studies based on the recent observation data of PM$_{2.5}$ (Zhang and Cao, 2015). The morning peak of PM$_{2.5}$ was used to attribute to the enhanced traffic emissions during morning rush hours (DeGaetano and Doherty, 2004). However, we found that the effect of traffic emissions was likely weakened as stricter emission standards were applied at recently years. Take Beijing for example, the Beijing Municipal Commission of Development and Reform (BMCDR) implemented National 3 vehicle emission standard at the beginning of December, 2005, and National 4 vehicle emission standard at March, 2008. Much tighter National 5 vehicle emission standard, equivalent to the Euro 5 emission standard was implemented at February, 2013 (www.bjpc.gov.cn). Based on the historical records of PM$_{2.5}$ data from our previous work (Liu et al., 2015), we calculated the diurnal variation of PM$_{2.5}$ during the four kinds of vehicle emission standard stages. As showed in Fig. 1, a visible morning peak of PM$_{2.5}$ was observed during the stages of National 2, 3 and 4 vehicle emission standard applied, but gradually disappeared or invisible after National 5 vehicle emission standard applied. As Shanghai and Guangzhou also implemented the corresponding National vehicle emission standard followed Beijing, the weakened effect of traffic emissions on PM$_{2.5}$ during the morning rush hours was predictable in these two megacities. For the other cities like Shenyang, Chongqing and Lhasa, however, the latest Nation vehicle emission standard was usually applied 2-3 years later than the megacities of Beijing, Shanghai and Guangzhou. Thus, the invisible morning peak of PM$_{2.5}$ was not observed at these three cities as they still applied National 4 vehicle emission standard during the observation period of this study. The related discussion have been added in the revised MS.

"The invisible morning peak of PM$_{2.5}$ in these three cities was possibly attributed to the stricter emission standards applied at recently years. As showed in Fig.S4, the morning peak of PM$_{2.5}$ in Beijing was gradually disappeared or invisible after National 5 vehicle emission standard applied at the beginning of 2013(www.bjpc.gov.cn). The same thing would be also observed in Shanghai and Guangzhou which implemented the same vehicle emission standards followed Beijing, while it not true for the other cities as the latest vehicle emission standard was usually applied 2-3 years later than the three megacities."

[Figure]

Fig.S5 Diurnal variation of PM$_{2.5}$ in Beijing during the four kinds of vehicle emission standard stages. PM$_{2.5}$ data obtained during Nation 2 vehicle emission stage (Nation 2) in this study refer to the periods of Jan. 2004 to Nov. 2005. Nation 3 refer to the periods of Dec. 2005 to Feb. 2008. Nation 4 refer to the periods of Mar. 2008 to Jan. 2013. Nation 5 refer to the periods of Feb. 2013 to Dec. 2014.

Zhao, X., Zhang, X., Xu, X., Xu, J., Meng, W., Pu, W. Seasonal and diurnal variations of ambient PM2.5 concentration in urban and rural environments in Beijing. Atmos. Environ. 43, 2893-2900, 2009.

Zhang, Y. L., and Cao, F. Fine particulate matter (PM$_{2.5}$) in China at a city level. Sci. Rep., 5: 14884. 2015.

DeGaetano, A. T., and Doherty, O. M. Temporal, spatial and meteorological variations in hourly PM2.5 concentration extremes in New York City. Atmos. Environ., 38, 1547-1558, 2004.

Liu, Z. R., Hu, B., Wang, L. L., Wu, F. K., Gao, W. K., and Wang, Y. S. Seasonal and diurnal variation in particulate matter (PM$_{10}$ and PM$_{2.5}$) at an urban site of Beijing: analyses from a 9-year study. Environ. Sci. Pollut. Res., 22, 627-642, 2015.

7. P9, line 360-361. The discussion of bimodal pattern of PM$_{2.5}$ in Lin'an is not convincing, more likely Lin'an is highly affected by the regional transportation from the YRD region, which was totally different from Namsto and Gongga Mountain.

[Response] We agree with the reviewer's comments, and revised the discussion of bimodal pattern of PM$_{2.5}$ in Lin'an.

"Both Gongga Mountain and Lin'an showed the same bimodal pattern of PM$_{2.5}$ as that in Namsto, the former site could also be influenced by the planetary boundary layer, while the latter site was not only influenced by the evolution of the planetary boundary layer but also would be highly affected by the regional transportation from the YRD region."

8. P10, line 392-394, need reference here for the calculate method of Si.

[Response] Thanks for pointing out this. One reference was cited here.

Mason, B.: Principles of Geochemistry, New York, Wiley, 1966.

9. P11, line 432-437, the explanation is not convincing, the higher fraction of sulfate in south China is more likely associated to the higher oxidation capacity in south China and therefore higher formation efficiency from $SO_2$ to $SO_4^{2-}$.

[Response] Thanks for pointing out this. We added this discussion in the revised MS.

"In addition, the higher fraction of sulfate in south China is also likely associated to the higher oxidation capacity in south China and therefore higher formation efficiency from $SO_2$ to $SO_4^{2-}$."

10. P11, line 456-460, what about the fraction of EC? Is it higher or lower compared with previous studies? More discussion about the comparisons with previous studies would be useful, at least for the urban sites.

[Response] Thanks for pointing out this. More discussion about the comparisons of EC with previous studies was provided.

"In addition, the EC fraction (5.7%) was slightly lower than those found in previous studies (7%-7.4%) (Yang et al., 2011; Wang et al., 2015a)."

Wang, H. B., Tian, M., Li, X., Chang, Q., Cao, J., Yang, F., Ma, Y., He, K.: Chemical Composition and Light Extinction Contribution of $PM_{2.5}$ in Urban Beijing for a 1-Year Period. Aerosol and Air Quality Research, 15, 2200-2211, 2015a.

Yang, F., Tan, J., Zhao, Q., Du, Z., He, K., Ma, Y., Duan, F., and Chen, G.: Characteristics of $PM_{2.5}$ speciation in representative megacities and across China. Atmos. Chem. Phys., 11(11), 5207-5219, 2011.

11. P16, better to split section 3.3 into two sub-sections. One section focuses on the mass concentration of $PM_{2.5}$ and the other on the chemical composition of $PM_{2.5}$.

[Response] Thanks for the suggestion. We divide section 3.3 into two sub-sections, and more discussion about the chemical evolution in the polluted days and the possible primary sources contributed to the high $PM_{2.5}$ loading in the specific regions were explored.

12. P17, line 722-724, "These results suggest the different formation mechanisms of the heavy pollution in the most polluted city clusters, and unique mitigation measures should be developed for the different regions of China." This conclusion is ambiguous, the authors should clearly state the difference in formation mechanisms, and the implications for mitigation measures.

[Response] Thanks for the suggestion. We admitted that conclusion is ambiguous. To make it clear, detailed description of chemical evolution on fine particles in each area was provided in this part and the implications for mitigation measures were drawn in the Conclusion.

"The increasing contribution of secondary aerosol on polluted days was observed both for the urban and nearby background sites, suggesting fine particle pollution in the most polluted areas of China assumes a regional tendency, and the importance to address the emission reduction of secondary aerosol precursors. In addition, the chemical species dominating the evolutions of the heavily polluted events were different, while decreasing or constantly contribution of OM associated with increasing contribution of SIA characteristic evolution of $PM_{2.5}$ in NCP, PRD and SWCR, the opposite phenomenon was observed in NECR. Further analysis from the $[NO_3^-]/[SO_4^{2-}]$ ratio and

OC/EC ratio showed that fine particle pollution in Guangzhou and Shenyang was mainly attributed to the traffic emissions and coal combustion, respectively, while more complex and variable major sources including mobile vehicle emission, biomass burning and coal combustion contributed to the development of heavily polluted days in Beijing. As for Chongqing, the higher oxidation capacity than other cities suggested it should pay more attention to the emission reduction of secondary aerosol precursors. These results suggest the different formation mechanisms of the heavy pollution in the most polluted city clusters, and unique mitigation measures should be developed for the different regions of China"

13. P26, Table 2, please provide the standard deviation of the concentration of $PM_{2.5}$ and its components.
[Response] Thanks for the suggestion. The standard deviation was added.

14. P31, Fig.5, need a clear figure caption.
[Response] Thanks for pointing out this. The caption of Fig. 5 was improved.

[revised manuscript text omitted]